# IMPLICIT STATE ESTIMATION VIA VIDEO REPLANNING

## ABSTRACT

Video-based representations have gained prominence in planning and decision-making due to their ability to encode rich spatiotemporal dynamics and geometric relationships. These representations enable flexible and generalizable solutions for complex tasks such as object manipulation and navigation. However, existing video planning frameworks often struggle to adapt to failures at interaction time due to their inability to reason about uncertainties in partially observed environments. To overcome these limitations, we introduce a novel framework that integrates interaction-time data into the planning process. Our approach updates model parameters online and filters out previously failed plans during generation. This enables *implicit state estimation*, allowing the system to adapt dynamically without explicitly modeling unknown state variables. We evaluate our framework through extensive experiments on a novel simulated manipulation task set, demonstrating its ability to improve replanning performance and advance the field of video-based decision-making.

## 1 INTRODUCTION

Learning from videos has gained significant traction in decision-making, as videos capture rich visual and dynamic information while aligning with how humans acquire knowledge. These properties make them a powerful medium for specifying tasks and learning diverse skills across contexts. Recent work has shown the effectiveness of video-based frameworks in enabling robots to learn behaviors such as object manipulation (Li et al., 2024) and navigation (Zhang et al., 2024), highlighting the value of video as a flexible and expressive representation.

This paper focuses on video as a planning representation. Given a goal and current observation, video planning systems generate imagined task executions and convert them into robot actions. Unlike symbolic or latent representations, videos naturally encode both perceptual and action information and generalize across tasks and environments. Prior works (Chang et al., 2020; Du et al., 2024a;b) leverage these properties to train universal agents using video-based predictions.

Despite promising results, existing video planning frameworks suffer from a crucial limitation: they lack mechanisms to integrate past interactions with the environment and cannot effectively reason about uncertainty due to partial observability. Consider the task of opening a door without knowing whether it should be pushed or pulled. If pushing fails, a human will naturally infer that pulling is the correct action and adjust accordingly. In contrast, current video planning frameworks simply generate a new plan, disregarding the knowledge gained from prior attempts. This inability to incorporate feedback from interactions significantly limits their effectiveness, particularly in unstructured environments where uncertainty is inherent.

A common solution introduces explicit belief models trained in simulation to infer hidden parameters (Memmel et al., 2024; Qi et al., 2023). Yet these methods require prior knowledge of relevant parameters and may struggle when interaction data fails to disambiguate uncertainties. In contrast, humans can effectively resolve these challenges by drawing on past experiences and employing adaptive trial-and-error strategies when prior knowledge is lacking. In the context of video planning, failed interactions are not merely setbacks — they offer critical information about the environment and system parameters.

Our goal is to develop a video planning framework that effectively encodes and retrieves interaction data, adapting plans dynamically based on past experiences—without requiring additional simulation data or prior knowledge of the parameters. We propose a novel problem formulation that formalizes the challenge of generating video plans while implicitly incorporating past interactions, as illustrated in Figure 1. To address this, we introduce implicit state estimation (ISE), a framework that integrates interaction-time data into the planning process without explicitly defined system pa-

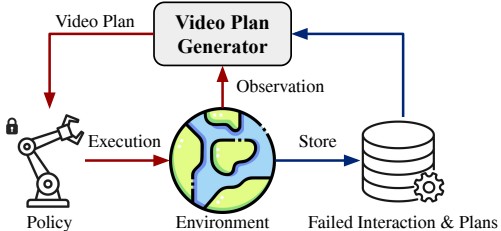

Figure 1: **Implicit state estimation via video replanning.** Previous video planning methods rely on the first frame (in red). We propose to incorporate past failures (in blue) to improve future plans.

rameters. Instead of relying on hand-specified beliefs, our approach enables rapid adaptation by optimizing a subset of model parameters that *implicitly* encode unobservable state features. ISE also incorporates a plan-rejection mechanism that filters out previously failed plans, preventing over-reliance on incorrect inferences and encouraging exploration. By adopting a video planning framework, our approach leverages the latest advancements in generative models, paving the way for future applications such as out-of-distribution detection.

To validate the effectiveness of our framework, we evaluate our approach on the Meta-World System Identification Suite, a novel task set for the Meta-World (Yu et al., 2019) environment designed to evaluate online adaptation in decision-making across diverse robotic tasks with unknown system parameters. Experiments show that our method allows for rapid adaptation without explicit system identification. Compared to baselines, including reinforcement learning methods and other methods for video planning, our approach significantly reduces replanning failures and generates more accurate video plans in dynamic, uncertain environments.

## 2 RELATED WORKS

**Video planning.** Videos have emerged as a popular medium for planning in the physical world (Yang et al., 2024b). In this work, we use the term "planning" to refer to synthesizing a continuous, goal-conditioned trajectory in visual space that can be directly executed or used to guide policy learning. This is distinct from symbolic action sequence search in classical AI planning or explicit policy search in POMDP solvers. Existing approaches have explored using video to train dynamics models (Yang et al., 2024a; Ajay et al., 2023; Zhou et al., 2024; Qin et al., 2024) or reward models (Ma et al., 2023; Chen et al., 2021a; Escontrela et al., 2023). Another popular approach involves directly predicting video sequence (Du et al., 2024a; Ko et al., 2024; Black et al., 2024) or derived representations such as 3D-correspondence or point tracks to guide a downstream policy (Yuan et al., 2024; Wen et al., 2024; Bharadhwaj et al., 2025). However, most of these frameworks do not consider environmental uncertainty, which limits their ability to efficiently adapt and replan, often resulting in suboptimal performance in such environments.

**System identification.** System identification focuses on learning system dynamics and estimating parameters using experimental data, with early work on input selection and the Fisher information matrix (Schön et al., 2011; Ljung, 1998; Menda et al., 2020). This research laid the groundwork for active identification strategies, which aim to maximize the amount of information gained from system inputs, a key element in classical system identification (Hjalmarsson et al., 1996; Lindqvist & Hjalmarsson, 2001; Gerencser & Hjalmarsson, 2005). Recent advances in the field have applied these methods to real-world systems, such as identifying physical parameters (Xu et al., 2019; Kumar et al., 2019; Mavrakis et al., 2020; Gao et al., 2020; 2022; Memmel et al., 2024). Concurrently, Li et al. (2025) has explored verifier-based frameworks that reject infeasible action proposals based on historical input, enabling online adaptation. Our work can be seen as bridging classical system identification and modern video planning method by injecting the ability to implicitly infer physical parameters from past video data to the model.

**Few-shot adaptation and latent context inference.** A widely adopted abstraction for few-shot adaptation models task variation through latent context or hidden system parameters. Hidden-Parameter MDPs (Doshi-Velez & Konidaris, 2016; Killian et al., 2017) formalize environment variation using unobserved parameters that must be inferred from interaction data. This perspective

has been extended in meta-reinforcement learning through latent context representations that are inferred from experience and used to condition policies (Rakelly et al., 2019; Sämundsson et al., 2018). More recent work has pursued scalable and robust latent-context inference, explicitly accounting for uncertainty in few-shot settings (Zhou et al., 2025; Haldar et al., 2023). In robotics, several methods study adaptation under dynamics shifts and embodiment variation, as well as adaptation to changes in physical properties (Kumar et al., 2021; Cho et al., 2024; Lee et al., 2023). In parallel, retrieval-based approaches reuse past interactions to guide specialization through sub-trajectory reuse (Memmel et al., 2025). Our approach also treats system variation as latent, but instead of maintaining beliefs for tuning policies, we optimize beliefs directly for test-time planning.

**Learning from offline data.** Imitation learning and offline reinforcement learning study how to leverage pre-collected datasets, including expert demonstrations and logged experience, to learn control policies (Zare et al., 2024; Ho & Ermon, 2016; Florence et al., 2021; Levine et al., 2020; Figueiredo Prudencio et al., 2024; Kidambi et al., 2020; Yu et al., 2020). Recent work has further expanded these paradigms using expressive sequence models and diffusion-based policies (Lai et al., 2024; Huang et al., 2024; Yeh et al., 2025; Chen et al., 2021b). While we also learn from offline data, we focus on extracting representations of interaction dynamics from videos, including failed trials, rather than directly optimizing policies from state-action or state-action-reward tuples.

## 3 PROBLEM FORMULATION

We consider a setting where certain system or task parameters $\theta$, such as mass, friction, or utility — essential for solving the task — are unknown. Unlike existing work for explicit state estimation, we do not assume access to the parameterization of $\theta$. We only assume that the underlying $\theta$ can be inferred from interaction data. For example, in a bar-pushing task, the center of mass can be implicitly inferred from an interaction video, regardless of whether the push is successful.

We aim to design a system capable of rapidly solving tasks by planning efficiently and leveraging interactions in the environment. For each environment, we assume access to an experience dataset $\mathcal{D}$, consisting of data tuples $d_i = (v_i, o_i, s_i) \in \mathcal{D}$, where $v_i$ represents an interaction video, $o_i$ denotes the object ID that indexes the underlying physical configuration during data collection, and $s_i$ is a binary success indicator. The dataset contains both successful interactions where $s_i = 1$ and failed interactions where $s_i = 0$. Our work focuses on evaluating whether the selected plans can guide successful action sequences. We enable this evaluation by using a fixed action prediction module to convert videos into executable actions.

## 4 APPROACH

Our world is full of uncertainty. For example, when trying to move a heavy piece of furniture, you often do not know its exact weight, how slippery the floor might be, or whether its legs will flex under pressure. As humans, we actively make new plans and use interaction trials to adapt: infer hidden parameters and narrow the space of possible dynamics.

Our framework follows this intuition. In the video planning framework, we propose to leverage prior interaction videos to refine the agent's belief about the environment through trial-and-error and video-based planning. Shown in Figure 2, given the current scene and past interactions, it retrieves and updates a latent embedding that implicitly captures the structure of the hidden parameters, then generates a video plan conditioned on this embedding.

Concretely, our video replanning framework maintains two test-time buffers: one for failed interactions and another for failed plans. At each round, it maintains an internal latent embedding that is updated based on observed failures, then generates new candidate plans guided by this latent embedding (Section 4.1). A rejection strategy (Section 4.2) selects the plan most different from past failures. The chosen plan is then converted into executable actions via an Action Module (Section 4.3), which can be instantiated with inverse dynamics models, goal-conditioned policies, or trajectory-tracking controllers. In our implementation, we adopt the training-free point-tracking method from AVDC (Ko et al., 2024). For extensive details of each component, please refer to Appendix C.

### 4.1 VIDEO-BASED PLANNING WITH STATE EMBEDDINGS

To plan effectively in environments with unknown dynamics, our system constructs a latent representation of hidden parameters, such as mass or friction, through *state embeddings*. These embed-

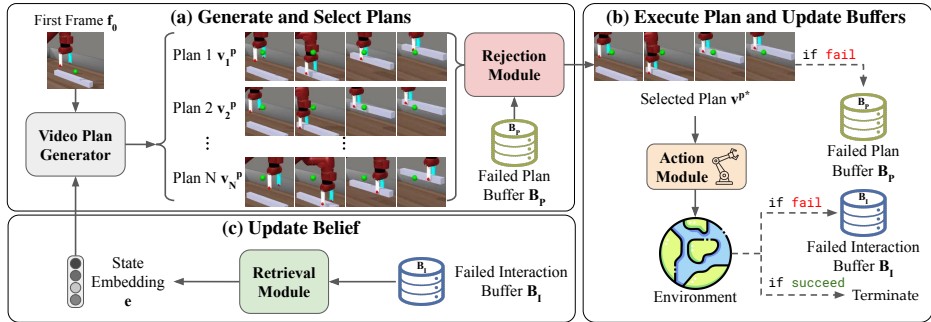

Figure 2: **Framework overview. (a) Generate and select plans** The Video Plan Generator first generates a set of candidate video plans conditioned on first-frame $f_0$ and state embedding $e$. The Rejection Module then selects the plan that is least similar to the plans in the Failed Plan Buffer $B_P$. **(b) Execute plan and update buffers.** The Action Module interacts with the environment by following the selected plan $v^{p*}$. If the interaction trial is not successful, we update the Failed Plan Buffer $B_P$ and the Failed Interaction Buffer $B_I$. **(c) Update belief.** The state embedding $e$ is updated using the Retrieval Module, which executes our **retrieval** and **refinement** procedures based on the Failed Interaction Buffer $B_I$. Then, we generate the next batch of video plans according to the updated belief.

dings act as hypotheses about the environment configuration and serve as conditioning signals for generating scenario-specific rollouts. By combining prior retrieval with test-time refinement, our framework maintains *state embeddings* as a flexible and adaptive representation of the environment.

**Plans vs. interactions.** We distinguish between *plans* and *interactions* because they originate from fundamentally different sources and serve different roles in our framework, even though both are represented as videos.

A **plan** is generated by the video planner and represents a hypothetical future trajectory based on current observation. Because the planner is trained only on successful executions, it tends to generate videos that resemble successful behavior. However, when its internal representation of the task or system parameters is inaccurate, it may produce implausible or physically incorrect plans (e.g., generating a video of pushing a door that should be pulled instead). Such inconsistencies typically show during execution, for instance, when the door remains stuck.

In contrast, an **interaction** is a real execution obtained directly from the environment. Interactions therefore reflect the true underlying system, although they may be noisy or visually ambiguous. Unlike plans, interactions are never physically incorrect in the sense of violating the actual dynamics of the environment.

**Representing state embeddings.** A *state embedding* is a single vector associated with an object instance, summarizing its underlying physical properties (e.g., mass, friction, or functionality) as revealed through interaction dynamics. We construct these per-object state embeddings in two stages: (i) encode every interaction video into a latent feature vector, and (ii) select one representative embedding per object ID.

*Stage 1: encoding interaction videos.* Let the dataset be $\mathcal{D} = \{(v_i, o_i)\}_{i=1}^N$, where each video $v_i$ is associated with an object ID $o_i$. We encode every interaction video using a pretrained vision encoder $E$, producing video-level embeddings $e_i = E(v_i)$ for all $i = 1, \ldots, N$. Interaction videos capture how the object responds to applied actions, so the resulting embeddings inherently reflect information about latent physical properties.

*Stage 2: constructing per-object state embeddings.* To obtain a single representation for each object ID, we group videos in $\mathcal{D}$ by their object ID. For each object ID $o$, we select one *successful* trial indexed by $\iota$ and define the corresponding *canonical embedding* as $e^o = e_\iota$. These canonical embeddings serve as compact descriptors of interaction dynamics and are used by our retrieval and planning modules.

**Context encoder $E$.** Given the experience dataset $\mathcal{D}$, the context encoder $E$ determines the feature space that the Retrieval Module and the Video Plan Generator operate on. To implement $E$, we experiment with pre-trained vision encoders known to offer strong semantic representations, such

as CLIP (Radford et al., 2021) and DINOv2 (Oquab et al., 2024). To encode videos with these image encoders, we uniformly subsample $F=8$ frames from the input video and encode each frame independently, then concatenate the resulting per-frame features to obtain a single-vector video representation.

**Retrieval and refinement.** At inference time, the planner requires a canonical embedding aligned with the current interaction. To obtain such an embedding, we retrieve candidates from prior interactions by comparing the features of the currently observed interaction video against stored embeddings. Instead of deterministically selecting the closest match, we treat the distances as similarity scores, normalize them with a softmax function, and sample a state embedding according to the resulting probability distribution. This stochastic retrieval helps the planner remain robust when multiple past interactions are plausible. Given an incoming interaction video $v$ with feature $e^v$, we compute distances

$$d_i = \|e^v - e^{v_i}\|_2^2, \tag{1}$$

and form a retrieval distribution

$$p_i = \frac{\exp(-d_i/\tau)}{\sum_j \exp(-d_j/\tau)}. \tag{2}$$

However, retrieval alone can still misalign with the ongoing interaction. To address this, we introduce an optimization-based refinement step illustrated in Figure 3: a generative identification module, which shares parameters with the *Video Plan Generator*, is trained to produce both successful and unsuccessful outcomes. At inference time, we freeze the module's weights and optimize the embedding to minimize the discrepancy between its generated rollouts and the observed interaction using a standard denoising diffusion loss. Concretely, we sample an initial embedding $e^{(0)} = e^{o_i}$ with $i \sim p$, and refine it using

$$e^{(k+1)} = e^{(k)} - \eta \nabla_e \mathcal{L}_{\text{refine}}(e^{(k)}; v), \tag{3}$$

$$\mathcal{L}_{\text{refine}}(e; v) = \mathbb{E}_{t,\epsilon}\left[\|\epsilon - \epsilon_\theta(x_t(v), t, e)\|_2^2\right], \tag{4}$$

where $\epsilon_\theta$ denotes the frozen generative identification model shared with the Video Plan Generator.

This refinement progressively adjusts the retrieved embedding toward the true environment dynamics, providing a more faithful basis for planning. While our full pipeline involves both retrieval and refinement processes, the refinement procedure can operate without initializing from a retrieved state embedding. In this case, we initialize $e^{(0)}$ by sampling from a multivariate Gaussian prior with per-dimension mean and variance estimated from all stored canonical embeddings:

$$e^{(0)} \sim \mathcal{N}(\mu, \Sigma).$$

We report the performance of this *refine-from-scratch* setting in the main experiments as a baseline.

**Video plan generation.** Once a state embedding is obtained and converted into a canonical embedding, the *Video Plan Generator* uses it to predict a sequence of plausible futures conditioned on the current observation. We adopt a diffusion-based planner following AVDC (Ko et al., 2024), but replace language-based conditioning with state embeddings. Specifically, the embedding is projected into the token space of CLIP-Text and appended as an additional conditioning token to the 3D U-Net backbone. The generator is trained end-to-end with a denoising diffusion objective, which encourages realistic rollouts under the hypothesized environment. In practice, we generate $M=7$ future frames at $128\times128$ resolution and concatenate them with the initial observation to form an 8-frame video plan. These plans are

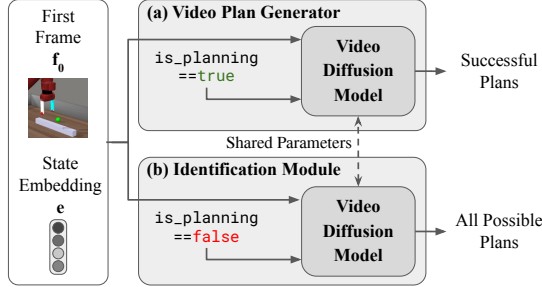

Figure 3: **Video Plan Generator and Identification Module.** We implement the Video Plan Generator and the Identification Module as two video generative models with shared parameters, switched with a binary trigger `is_planning`. The Video Plan Generator only learns from successful videos in the experience dataset $\mathcal{D}$, while the Identification Module is trained with all videos in $\mathcal{D}$, including the failures, enabling the state-embedding refining process.

then used to guide the action module throughout the next episode. The plans capture not only the high-level task intent but also environment-specific details encoded in the embedding, making them a suitable substrate for downstream rejection and action execution.

## 4.2 REJECTION-BASED REPLANNING

While the retrieval and refinement procedure introduced in Section 4.1 provides a plausible hypothesis of the environment, it is rarely perfect, especially with limited interaction data or in previously unseen situations. To improve robustness, we introduce a *rejection module* that actively explores alternatives, encouraging the planner to avoid repeating failed strategies.

**Candidate plan sampling.** At each planning round, instead of generating a single rollout, the Video Plan Generator produces $N$ candidate plans $\{v_1^p, v_2^p, \ldots, v_N^p\}$ by conditioning on $N$ independently sampled embeddings from the Retrieval Module. Because each sampled embedding represents a different hypothesis of the environment parameters, the resulting video plans can differ substantially, covering a range of possible outcomes.

**Rejection strategy.** To ensure the planner does not simply repeat failed behavior, we maintain a buffer $\mathcal{F}$ of previously failed plans. For each candidate plan $v_i^p$, we compute its distance to the closest failed plan in $\mathcal{F}$:

$$d(v_i^p, \mathcal{F}) = \min_{F \in \mathcal{F}} K(v_i^p, F),$$

where $K$ denotes the distance function. Empirically, we found that using an $L2$ distance directly on the pixel space of video frames works well. The rejection module then selects the plan with the largest distance to past failures,

$$v^{p*} = \arg\max_{v_i^p} d(v_i^p, \mathcal{F}),$$

which is subsequently passed to the Action Module for execution. This procedure encourages the planner to explore novel strategies rather than repeating unsuccessful ones, allowing the system to refine its behavior through active interaction.

## 4.3 ACTION MODULE

The final step is to convert a selected video plan into executable low-level actions for the robot. We follow the idea of dense object- or wrist-tracking from AVDC (Ko et al., 2024): the predicted video frames are used to track either the object of interest or, in tasks such as bar picking, the robot's wrist when object tracking is insufficient. The resulting trajectory is then translated into control commands through simple heuristic policies such as grasp strategies or trajectory-following controllers. This training-free approach enables direct execution of video plans without requiring an additional learned policy.

## 5 EXPERIMENT

### 5.1 META-WORLD SYSTEM IDENTIFICATION SUITE

Existing manipulation benchmarks primarily emphasize single-episode task completion while assuming full observability, overlooking the rich information available through interaction. Consequently, they fail to capture challenges posed by environmental variation, such as determining whether a door should be pushed or pulled, that humans frequently encounter in the real world and can solve with ease. Therefore, in this work, we propose the Meta-World System Identification Suite to evaluate online adaptation under unknown system parameters $\theta$. It combines an offline experience dataset with environments exhibiting hidden parameters that agents must infer through interaction. All environments act in the same 4-dimensional positional control action space $(dx, dy, dz, g)$, where the first three components are end-effector displacements and $g$ controls the gripper. Table 1 shows the set of

Table 1: **Meta-World System Identification Benchmark.** Each task involves interaction-based inference of system parameters $\theta$, which are either continuous or discrete with varying mode counts.

| Task | Identification Type |
|------|---------------------|
| Push Bar | Continuous |
| Pick Bar | Continuous |
| Slide Brick | Continuous |
| Open Box | Discrete, 2 modes |
| Turn Faucet | Discrete, 2 modes |

Figure 4: **Qualitative results of adapted video plans.** We show qualitative examples of adaptation trials for each task in the Meta-World Identification Suite to illustrate the effectiveness of the proposed method. The framework generates new plans based on previously failed interaction and video plans.

5 tasks, including object manipulation (e.g., push bar, pick bar) and puzzle solving (e.g., turn faucet), test adaptability to randomized centers of mass, uncertain interaction modes, and external forces.

**Push bar.** The objective of this task is to push a bar toward a designated target with the center of mass of the bar randomized.

**Pick bar.** This task requires the robot to grasp a bar and transport it to a specified target. Similarly, the center of mass of the bar is randomly assigned.

**Slide brick.** This task consists of two distinct stages. In the first stage, the robot must push a brick up an inclined surface. In the second stage, the brick slides freely down the slope. The objective is to control the initial push such that the brick comes to rest within a predefined target region on the slope.

**Open box.** The objective of this task is to open a box. The box cover can be manipulated in one of two modes: it can either be lifted or slid open.

**Turn faucet.** In this task, the robot must rotate a faucet. The faucet's direction of rotation is either clockwise or counterclockwise.

In the above five tasks, the system parameters cannot be inferred without physical interaction. The details of the tasks are provided in Appendix B.

## 5.2 EVALUATION METRICS

Our study focuses on two key aspects. First, we hypothesize that our replanning mechanism can reduce the number of replans required for successful execution. To quantitatively measure the rate of adaptation, we report the average number of replans needed per successful attempt in our main experiment. Specifically, the agent must repeatedly replan until it either succeeds or reaches a predefined maximum number of replan trials, $M$. For all tasks, we set $M = 14$.

Second, instead of only evaluating task success, we also assess the effectiveness of the replanning mechanism by comparing the replanned videos to ground-truth interaction videos. To quantify this similarity, we use standard video comparison metrics: PSNR, LPIPS (Zhang et al., 2018), CLIP-score, and DINO-score.

## 5.3 IMPLEMENTATION AND BASELINES

We evaluate our framework on the 5 tasks in the Meta-World System Identification Suite. For our model, the Video Plan Generator is trained on the videos in the offline experience dataset as described in 4.1. Each task is evaluated over 400 trials. We compare our approach with the original AVDC implementation for video planning, which does not online adapt, variants of our method that use different context encoders $E$, and other reinforcement learning baselines.

Table 2: **Number of replans until success.** Video planning methods, AVDC and ours, outperform baselines in imitation learning (BC) and offline RL (CQL) on most tasks. Our method and its variant consistently outperform the video planning baseline. The error terms show the standard error of the mean value. The reported error terms represent the standard error of the mean across 400 trials. For the CQL baseline, performance is averaged over three independently initialized and trained policies for more robust evaluation.

| Method | Push Bar | Pick Bar | Slide Brick | Open Box | Turn Faucet | All (Normalized) |
|---|---|---|---|---|---|---|
| CQL | 9.96± 0.20 | 9.22± 0.19 | 8.94± 0.18 | 3.32 ± 0.18 | 5.63 ± 0.21 | 2.56± 0.10 |
| BC | 10.26±0.30 | 8.79±0.32 | 9.24±0.33 | 1.59± 0.19 | 3.54± 0.27 | 1.98± 0.05 |
| AVDC | 6.83± 0.27 | 4.41± 0.22 | 7.36± 0.27 | 1.82± 0.16 | 1.67± 0.16 | 1.31± 0.04 |
| Ours (Refine) | 4.66±0.23 | 3.87±0.21 | 6.72±0.26 | 1.73±0.16 | 1.54±0.15 | 1.12± 0.03 |
| Ours | **4.26**± 0.20 | **3.84** ± 0.18 | **6.69** ± 0.26 | **1.25**± 0.10 | **1.39**± 0.14 | **1.00** ± 0.03 |

- **AVDC (Ko et al., 2024).** The baseline with only the Action Module, *i.e.*, , without the Retrieval Module and Rejection Module introduced in this paper.

- **Ours.** Our proposed method with $E$ based on DINOv2 feature as described in 4.1.

- **Ours (Refine).** A variant of our method with refining state embeddings from scratch as mentioned in 4.1.

Our problem formulation for state estimation can be viewed as a single-step multi-task reinforcement learning (RL) problem, where the success signal $s_i$ from the dataset serves as the reward, by additionally assuming privileged access to action labels (*i.e.*, ground-truth system parameters). We also implement such explicit state estimation baselines with popular RL algorithms to calibrate the difficulty of the tasks.

- **BC** uses a state embedding-conditioned behavior cloning framework where the policy takes in state and state embedding directly. The state embedding is obtained with $E$ based on DINOv2.

- **CQL** (Kumar et al., 2020) is a state embedding-conditioned actor-critic framework. The Q-function takes in state, action, and state embedding, while the Policy takes in state and state embedding, which is obtained with $E$ based on DINOv2.

## 5.4 RESULTS

The replanning efficiency of all methods is presented in Table 2. Our method and its variants consistently outperform baselines in video planning (AVDC), imitation learning (BC), and offline RL (CQL). AVDC performs better than BC and CQL in terms of trial efficiency, demonstrating the benefits of video-based planning. BC and CQL struggle significantly on these tasks and often fail to complete them, frequently repeating similar predictions—suggesting overfitting or limited flexibility. Overall, our approach achieves the strongest performance, showing its broad applicability across both discrete and continuous parameter settings.

**Our variants.** Among the variations of our method, using the Rejection Module (Ours) alone to generate state embeddings achieves the best overall performance. In contrast, the refinement process introduced in Section 4.1 alone, Ours (Refine), yields similar or slightly worse performance but *does not assume data availability at test time*, *i.e.*, do not require access to the dataset used to train the diffusion model. Additionally, its formulation as a learned model offers the potential for

Table 3: **Evaluation on replanned videos.** We quantitatively assess the accuracy of the generated video by measuring the similarity between the framework's output plan and the ground-truth plan produced by the GT scripted policy.

| Method | PSNR ↑ | LPIPS ↓ | CLIP ↑ | DINO ↑ |
|---|---|---|---|---|
| AVDC | 19.426 | 0.271 | 0.870 | 0.705 |
| Ours (R3M) | 19.632 | 0.261 | 0.875 | 0.705 |
| Ours (CLIP) | 19.584 | 0.262 | 0.871 | 0.707 |
| Ours (DINOv2) | **19.817** | **0.255** | **0.882** | **0.717** |

improved generalization. Finally, we show qualitative results for adaptations in Figure 4.

**Video plan evaluation.** In Table 3, we report the evaluations of video plans generated by the baseline (AVDC) and our methods with various context encoders $E$, including R3M (Nair et al., 2022),

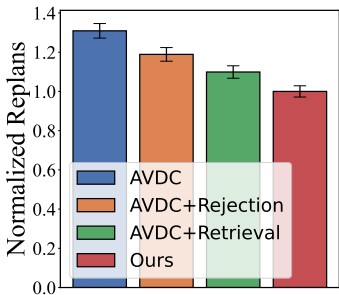

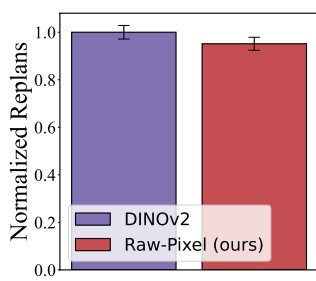

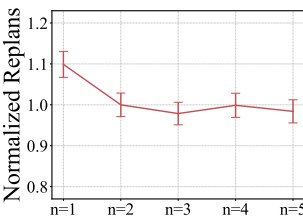

Figure 5: **Rejection and retrieval modules.** Both the Rejection and Retrieval Modules improve the performance from AVDC (Action Module only), and combining them (ours) yields the best performance. In this figure, the numbers of replans are normalized and aggregated among the five tasks.

Figure 6: **Rejection module distance.** We compare using raw-pixel distance and DINOv2 embedding distance to reject and the former yields better performance. The numbers of replans are normalized and aggregated among the five tasks.

Figure 7: **Number of generated video plans.** We experiment with generating different numbers of video plans $n \in [1, 5]$ and set $n = 2$. In this figure, the numbers of replans are normalized and aggregated among the five tasks.

CLIP (Radford et al., 2021), and DINOv2 (Oquab et al., 2024). Our method produces significantly higher-quality videos compared to AVDC. Our method with DINOv2 outperforms its counterpart using R3M or CLIP, validating our design choices.

## 5.5 Ablation studies

**Rejection and retrieval modules.** To investigate the effectiveness of the Rejection and Retrieval Modules, we compare AVDC (Action Module only), AVDC+Rejection, AVDC+Retrieval, and our method (with both Rejection and Retrieval). In AVDC, the Video Plan Generator is conditioned only on the current image observation $f_0$. AVDC+Retrieval conditions the generator on a single retrieved state embedding (with refinement) in addition to $f_0$, but does not use plan rejection. AVDC+Rejection samples multiple plans without any embedding and selects one using the Rejection Module. Our full method applies both retrieval and rejection. Notably, the AVDC+Rejection baseline is conceptually similar to a recent work (Li et al., 2025), which also uses a generator to sample candidate plans and a separate module to select the most viable one given previous interaction attempts. The result in Figure 5 shows the normalized numbers of replans across all the tasks, and individual task performance is shown in Appendix D. Both AVDC+Rejection and AVDC+Retrieval outperform AVDC, justifying the effectiveness of the Rejection and Retrieval modules. Our full method (with both the Rejection and Retrieval modules) achieves the best performance, validating the combinations.

**Rejection module distance metric.** We compare pixel-wise distance and DINOv2-based similarity for selecting video plan candidates, as described in Section 4.2. Figure 6 shows that pixel-wise distance performs better, likely because DINOv2 struggles to distinguish these plans due to the nature of its pretraining data.

**Number of generated video plans.** To study how the number of video plans generated to be rejected affects the performance, we vary the number from 1 to 5 and show the result in Figure 7. The result aggregates the performance across all the tasks. From $n = 1$ (AVDC+Retrieval) to $n = 2$ (ours), around 10% of replans are saved. However, from $n = 2$ to $n = 5$, the number of replans until success is nearly the same. An increase in video plan candidates generated by the video generator results in an increase in the computational cost. Therefore, since the performances of $n \in \{2, 3, 4, 5\}$ are comparable, we set $n = 2$.

Table 4: **Rejection module distance metric.** We compare the performance of using different distance functions for the Rejection Module.

| Metric | Push Bar | Pick Bar | Slide Brick | Open Box | Turn Faucet | All (Normalized) |
|---|---|---|---|---|---|---|
| Raw-Pixel (ours) | **4.26**± 0.20 | **3.83**± 0.18 | 6.69± 0.26 | **1.25**± 0.10 | **1.39**± 0.14 | **1.00**± 0.03 |
| DINOv2 | 4.83± 0.22 | 4.08± 0.20 | **6.62**± 0.26 | 1.27± 0.11 | 1.47± 0.14 | 1.05± 0.03 |

**Rejection module distance metric.** To study the impact of the distance function used in the Rejection Module, we evaluated several alternatives. The results show that raw-pixel distance consistently achieves superior or comparable performance relative to the DINOv2 baseline. Based on this finding, we adopt raw-pixel distance as the default choice of $K$ for the Rejection Module.

## 5.6 EVALUATION ON REAL-WORLD DATA

To demonstrate the method's ability to plan efficiently, we evaluated the video-planning framework on a real-world door-opening task. The door can be either pushed or pulled open, which can't be distinguished from the appearance and must be revealed through interaction. The setup is shown in Figure 8.

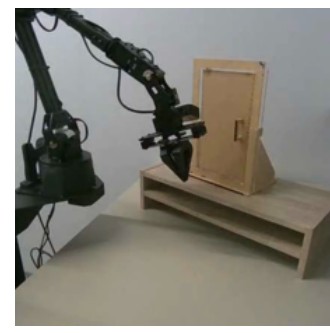

We collected 20 real-world videos (10 successes, 10 failures) to train the video model. We address the challenge of training on such a small dataset by adopting a joint training with simulation data: 40% of samples came from real-world data and 60% from the Meta-World System Identification Suite. To reduce overfitting to the first frames, which often leads to monotonic plans, we injected Gaussian noise into the first frames during inference. Furthermore, we set $n = 4$ to promote diverse plan generation.

For simplicity, we evaluate the single-episode replanning success rate, conditioned on an initial failure—different from the main evaluation metric. A perfect random policy, which assumes knowledge of the correct modes (e.g., push or pull) but does not use past interaction, would succeed with 50% probability. Our method achieved 15/20 successes, compared to 8/20 for a naive video planner without using past information, demonstrating the effectiveness of our pipeline.

Figure 8: **Real-world setup.** A WidowX arm attempts to open a door that can be pulled or pushed open.

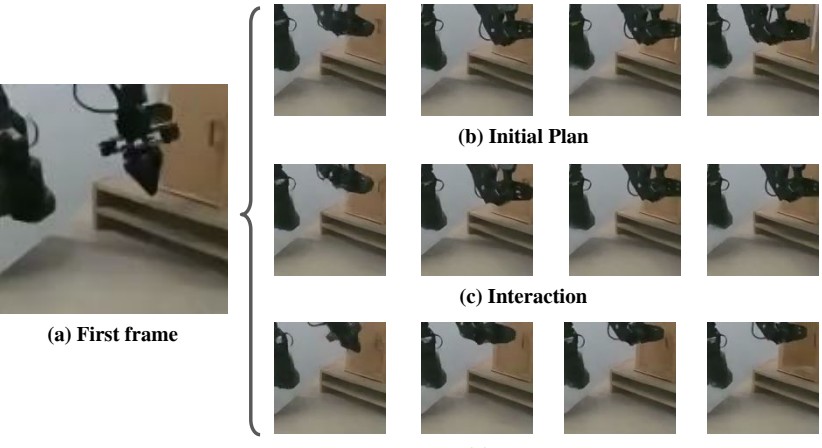

(a) First frame

(b) Initial Plan

(c) Interaction

(d) New Plan

Figure 9: **Real-world qualitative results.** Given the first frame (a), the robot first attempts to push the door (b). Then, it gets stuck and fails to open the door (c). Finally, it generates a plan that pushes the door instead (d) by leveraging information from previous failures.

Figure 9 shows the qualitative rollout for our real-world experiment. We followed most of the hyperparameters as used in the Meta-World System Identification Suite. Additionally, we changed $n = 4$ and injected Gaussian noise of standard deviation 0.35 into the first frames with pixel values normalized to $[0, 1]$ to mitigate the first-frame bias issue.

## REPRODUCIBILITY STATEMENT

We are committed to reproducibility and will release the full codebase for reproducing the experiments introduced in this work. Our experiments were conducted on standard GPUs (V100/3090)

with moderate training times, and we include full details in the appendix to facilitate independent verification.

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

APPENDIX

# Table of Contents

## A  SUPPLEMENTARY MATERIALS

### A.1  WEBSITE AND CODE

We present extensive qualitative results and release our code in our website. Our codebase can be found here.

## B  BENCHMARK DETAILS OF SIMULATOR TASKS

The environments of the five tasks are shown in Figure 10.

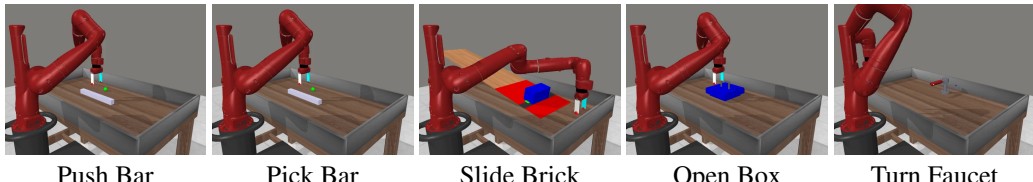

| Push Bar | Pick Bar | Slide Brick | Open Box | Turn Faucet |

Figure 10: Settings of five task environments.

## B.1 PUSH BAR

The objective of this task is to push a bar towards the target. Physically, the bar is combined with a mass point and a mass-free bar. Therefore, if the robot does not push the bar exactly at the center of mass, torque is imposed on the bar, resulting in the rotation of the bar.

The bar has a length of 0.4 meters. For ease of data collection, we discretize the center of mass position during pretraining. The data distribution of the center of mass on the bar is shown in Figure 11. During evaluation, the C.M is randomly sampled from $[-0.18, 0.18]$.

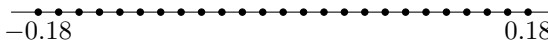

$$-0.18 \qquad\qquad\qquad\qquad 0.18$$

Figure 11: **Center of mass positions.** We visualize the distribution of the center of mass for our bar tasks. Each point corresponds to a possible position of the center of mass.

The visualization of this task is shown in Figure 12.

## B.2 PICK BAR

The objective of this task is to grasp a bar towards the target. Physically, like the push-bar mentioned above, the bar is combined with a mass point and a mass-free bar. Therefore, if the robot does not grasp the bar exactly at the center of the mass, torque is imposed on the bar, resulting in the tilt of the bar.

The properties of the bar in this task are the same as the bar in push-bar. And the distances between the center of mass and the geometric center of the bar in the pretraining stage are also the same as above.

The visualization of this task is shown in Figure 13.

## B.3 SLIDE BRICK

There are two stages in this task. In the first stage, the robot pushes a brick up an inclined surface. In the second stage, the brick slides freely down the slope. The objective is to push the brick at a proper height so that the brick can slide down and stop at the gray band.

Note that the wooden slope has a relatively low friction coefficient and that the red slope has a relatively high friction coefficient. Therefore, the brick can slide down and accelerate in the wooden slope and decelerate on the red slope. The friction coefficient of the wooden slope is fixed and is set to 0, while the friction coefficient of the red slope varies. During training, we discretize the friction coefficient as $\mu \in \{0.24, 0.25, 0.26, \ldots, 0.40\}$ in our dataset $\mathcal{D}$. During evaluation, the friction coefficient is sampled from $[0.24, 0.40]$.

The visualization of this task is shown in Figure 14.

## B.4 OPEN BOX

The objective of this task is to open a box whose lid operates in one of two possible ways: it either needs to be lifted or slid open. The specific mode is chosen at random and cannot be identified

visually, so the robot must physically interact with the box and adjust its strategy based on the observed response.

The visualization of the task is shown in Figure 15.

### B.5    TURN FAUCET

The objective of this task is to rotate a faucet. The direction of rotation—clockwise or counterclockwise—is randomly assigned and cannot be determined in advance through observation. The robot must interact with the faucet to identify the correct direction and execute the appropriate motion.

The visualization of the task is shown in Figure 16.

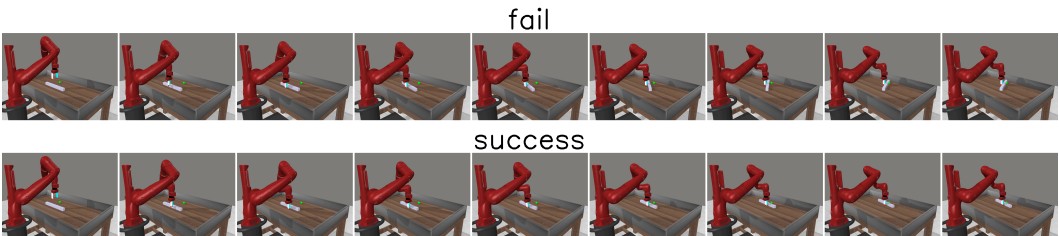

Figure 12: Push Bar. The above video shows a failed trial, because the deflection of the bar is too large so that the bar cannot reach the target.

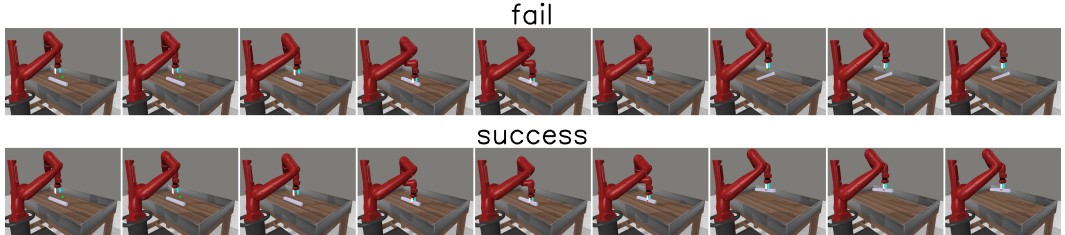

Figure 13: Pick Bar. The above video shows a failed trial, because the bar fell down.

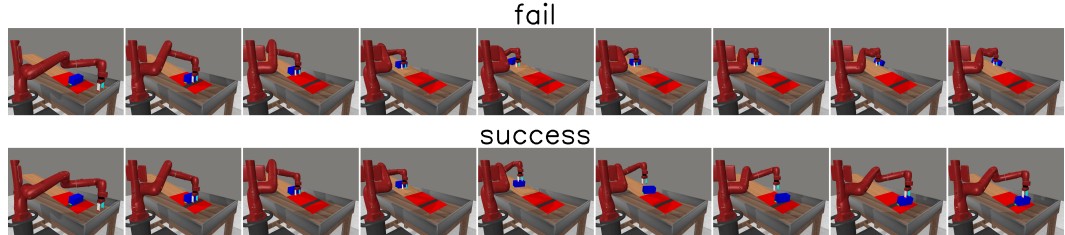

Figure 14: Slide Brick. The above video shows a failed trial, because the brick didn't slide down to the gray band.

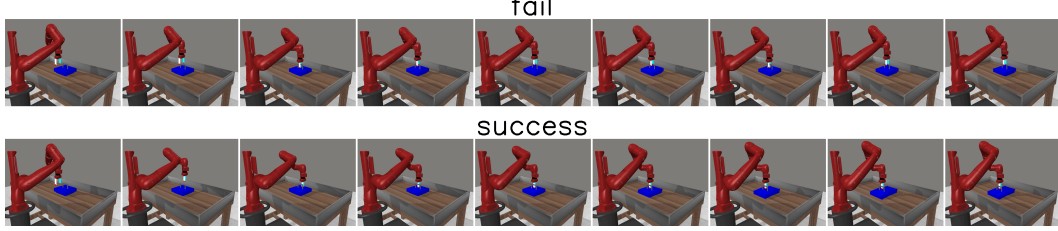

Figure 15: Open Box. The above video shows a failed trial, becuase the robot try to grasp the handle up. However, the right way to open the box is to push the handle horizontally.

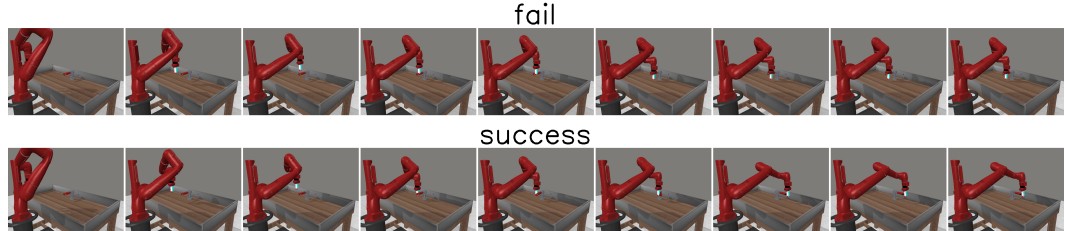

Figure 16: Turn Faucet. The above video shows a failed trial, because the robot try to turn the faucet clockwise. However, the right way is to turn the faucet counterclockwise.

Figure 17: Visualization of simulation tasks.

## C  METHOD DETAILS

### C.1  VIDEO PLAN GENERATOR AND RETRIEVAL MODULE

To generate meaningful plans in environments with unknown dynamics, our system first hypothesizes the hidden parameters of the environment via the retrieval and refinement of a latent state embedding, which serves as a proxy for the current environment configuration. We first discuss our *Video Plan Generator* and *Retrieval Module*, which generate candidate plans based on observed images and state embeddings. The *Video Plan Generator* predicts future frames conditioned on the current observation and, when available, the state embedding. For new objects without prior interactions, the generator operates solely on the observed image by conditioning on a learned "null" state embedding.

The *Retrieval Module* extracts state embeddings from prior interactions by encoding past videos and selecting a representative embedding for each object ID. At inference time, it encodes the current interaction video, computes distances to past embeddings from dataset $\mathcal{D}$, and uses a softmax function to sample the most relevant state embedding. In addition, we use an additional identification model to continuously refine the selected state embedding to best fit current interaction videos.

### C.1.1  STATE EMBEDDING

To represent the hidden system parameters, we encode interaction videos from previous action executions with the same object using a context encoder $E$. The video embedding for each video $v_j$ is obtained as $e_j^v = E(v_j)$. In the dataset $\mathcal{D}$, multiple data tuples may share the same object ID $o_j$. Naively encoding each video independently would result in multiple, different embeddings for the same object. To address this, we preprocess the dataset by grouping all entries with the same object ID and selecting the encoding from one of the successful executions (where $s_j = 1$) to represent the object, resulting in a state embedding $e_j^o$.

### C.1.2  VIDEO PLAN GENERATOR

The Video Plan Generator is used to generate a video containing $M$ future frames conditioned on the current observation and a state embedding. We follow the implementation of AVDC's video planner, which is a first-frame conditioned video diffusion model utilizing a 3D U-Net architecture. The original implementation conditions on task features were obtained from encoding natural language task descriptions with text encoders such as CLIP-Text, rather than state embeddings. To adapt the method to our needs, we project the state embedding into the hidden dimension of the language features and treat the projected embedding as an additional language token. The parameters of the projection layer are trained end-to-end along with the video planner, which is optimized using a standard denoising diffusion loss. For all experiments in this paper, we choose the resolution of generated frames to be 128x128 and $M = 7$. Each time we invoke the Video Plan Generator, $M = 7$ future frames are generated, we then concatenate them with the first frame, resulting in a $M + 1 = 8$ frames video plan.

Table 5: **Hyperparameters.** Comparison of configuration parameters for the Meta-World benchmark.

| | |
|---|---|
| **num_parameters** | 125M |
| **diffusion_resolution** | (32, 32) |
| **target_resolution** | (128, 128) |
| **base_channels** | 128 |
| **num_res_block** | 2 |
| **attention_resolutions** | (2, 4, 8) |
| **channel_mult** | (1, 2, 3, 4) |
| **batch_size** | 128 |
| **training_timesteps** | 12k |
| **denoising_timestep** | 100 |
| **sampling_timestep** | 25 |

Our video diffusion model adopts a U-Net architecture following (Ko et al., 2024). The original implementation conditions on the first frame and the task name tokens, where the task tokens are encoded using CLIP-Text, to predict future frames. Building on this framework, we additionally condition the video generation process on past interaction videos. Specifically, we encode the past interaction video as described in Appendix C.1.5, and concatenate the resulting embedding with the encoded task tokens after projecting it to match the CLIP-Text output dimensionality (512 channels) using a trainable linear layer.

To improve computational efficiency, the video diffusion model generates a low-resolution video plan through the denoising process. This low-resolution plan is then upsampled to the target resolution using a separate super-resolution model that predicts residuals over the upsampled frames. The diffusion model is trained using the v-parameterization approach with a learning rate of 1e-4. Key hyperparameters for the video diffusion model are summarized in Table 5.

### C.1.3 RETRIEVAL MODULE

During inference, the Video Plan Generator needs to take in a state embedding extracted from prior interactions. To ensure that the method is general across different encoders $E$, we first normalize and align the dimensionality of the feature space constructed by video features $e_j^v$ by applying PCA to all extracted video features $e_j^v$, yielding a PCA projection $P$ and PCA-transformed features $e_j^{vp} = P(e_j^v)$ and $e_j^{op} = P(e_j^o)$.

Given the interaction video from a prior execution, $v^e$, we first encode it using the encoder $E$ and apply the PCA transformation, resulting in $e^{ep} = P(E(v^e))$. We then compute the distance between $e^{ep}$ and the video encodings from $\mathcal{D}$ as $logits_j = -dist_j = -K(e^{ep}, e_j^{vp})$, where $K$ represents the distance function. For the choice of $K$, we used Cosine Distance for CLIP and DINO-based methods and L2 Distance for other methods. After calculating the logits, we compute the sampling probability of each output embedding by passing the logits through a softmax function: $p_j = \text{softmax}(logits_j/\tau)$, where $\tau$ is the temperature parameter. Finally, we sample the output state embedding $e = e_k^{op}$ according to the resulting probability distribution $k \sim \text{Categorical}(p_1, p_2, \ldots, p_n)$. While such retrieval-based state estimation formulation necessitates the data being available during test time, it enables training-free state estimation and guarantees that the Video Plan Generator never accepts out-of-distribution state embeddings during inference time.

### C.1.4 REFINING STATE EMBEDDING

In addition to using retrieval to find a relevant state embedding for a video, we also use an optimization-based approach to find a refined state embedding corresponding to a specific interaction video. Specifically, we introduce another generative model, referred to as the Identification model or the ID module, trained with the parameters shared with the Video Plan Generator, as shown in Figure 3. The ID module is trained to generate all possible interaction outcomes, including both successful and unsuccessful ones.

During replanning, we freeze its parameter and only optimize the state embedding to maximize the probability of generating the given interaction video.

$$e = \arg\min_e MSE(v^e, v)$$

Here, $e$ is the optimized state embedding, $v^e$ is the interaction video obtained at test-time, and $v$ is the videos sampled from the ID module conditioned on $e$.

Since the ID module and the Video Plan Generator share the same embedding space for state embeddings, the optimized state embedding can be used directly to generate video plans. In the experimental section of the paper, we analyze the effect of this refinement procedure for state estimation when 1) this approach optimizes state embedding from randomly initialized vectors drawn from unit variance Gaussian distribution, 2) the state embedding is set to the output of the Retrieval Module, and 3) these two approaches are combined, where this approach optimizes the output of the Retrieval Module.

### C.1.5 LATENT FEATURE SPACE

Given the experience dataset $D$, context encoder $E$ decides the feature space that the Retrieval Module and the Video Plan Generator operate on. To implement $E$, we experiment with taking pre-trained vision encoders that are known to give good semantic representations, such as CLIP (Radford et al., 2021) and DINOv2 (Oquab et al., 2024). To encode videos with these image encoders, we used a simple strategy that first encodes each frame independently and then concatenates the per-frame features, resulting in a single-vector video feature. A visualization of the constructed feature space is shown in Appendix D.1.

### C.2 REJECTION MODULE

While the modules introduced in Appendix C.1 alone are sufficient to perform system identification essential for solving the tasks given a set of videos of interactions in the environment, the identified system parameters are often not perfect. This is especially the case when we only have a limited number of interactions with the environment. Thus, to effectively estimate the precise state of the environment, it is important that the system repeatedly actively interacts with the environment.

To actively interact with the environment, we use a video plan generator to generate plans for execution. To encourage the planner to explore novel plans and avoid being overly reliant on suboptimal beliefs, we employed a simple rejection-based sampling method. In the Rejection Module, the previously failed plans are stored in a data buffer. At each planning round, we generate $N$ plans $(v_1^p, v_2^p, \ldots, v_N^p)$ instead of one by conditioning the Video Plan Generator on $N$ independently sampled state embeddings from the Retrieval Module, yielding $N$ different plans with potentially different beliefs on environment parameter $\theta$. Let $\mathcal{F}$ be the set of previously failed plans stored in the data buffer, we first calculate the distance to the nearest failed plan for each plan $P_i$:

$$d(v_i^p, \mathcal{F}) = \min_{F \in \mathcal{F}} K(v_i^p, F)$$

where $K$ is the distance function. Empirically, we found that L2 distance on raw pixel space works surprisingly well already.

We then select the plan $v^{p*}$ whose distance to its nearest failed plan is the largest (out of a set of generated video plans):

$$v^{p*} = \arg \max_{v_i^p} d(v_i^p, \mathcal{F})$$

The selected $v^{p*}$ is later converted to low-level control signals through the Action Module.

The combination of Appendix C.1 and Appendix C.2 form the core of our Implicit State Estimation procedure. Interactions with the environment through the rejection module in Appendix C.2 obtain a buffer of interaction videos that help describe the implicit state of the environment. The retrieval module in Appendix C.1 then obtains an explicit latent that represents the implicit state of the environment.

### C.3 ACTION MODULE

Given a generated video plan, to convert the video into continuous actions to execute, we follow the concept of dense object-tracking from AVDC to recover actions from video plans. On certain interaction tasks, the object of interest is not applicable. For example, in a bar-picking task where the robot is required to grasp around the center of mass to succeed, knowing the future trajectory of the bar is not sufficient to decide the grasp location. In these tasks, we track the robot's wrist instead. After obtaining the object/robot wrist trajectory from the tracking results, we convert the trajectory into actions using hand-crafted heuristic policies.

### C.4 DATA COLLECTION PROCESS

In simulation, we use scripted policies to collect offline interaction data. Specifically, the policy takes as input both the current observation and a "guessed" system parameter. For example, in

the Bar Picking task, the guess is a one-dimensional scalar representing the position of the center of mass relative to the bar. If this guess matches the true underlying parameter of the environment (*e.g.*, , both are centered), the interaction results in a successful video; otherwise, it produces a failed interaction. These outcomes correspond directly to the success signal $s_i$ described in Section 3.

For each object, we perform multiple interactions with different guesses, generating several failed trials, followed by an interaction using the correct guess, which produces a successful trial. We then assign all trials from the same object the same object ID $o_i$ and terminate the interaction loop for this object. The policy then starts interacting with a new object by randomizing the initial configuration and hidden parameters. As for the real-world experiment, we simply used tele-operation to replace the guess-augmented scripted policy.

# D ADDITIONAL EXPERIMENT RESULTS

## D.1 STATE EMBEDDING FEATURE SPACE

Figure 18 visualizes the feature space introduced in Section 4.1, using DINOv2 (Oquab et al., 2024) as the encoder. The plots are generated by applying t-SNE to sets of state embeddings from each environment, where for continuous modes we randomly sample four values for visualization. Our method yields a well-structured feature space in which different modes can be separated without additional data-processing, facilitating more effective downstream learning.

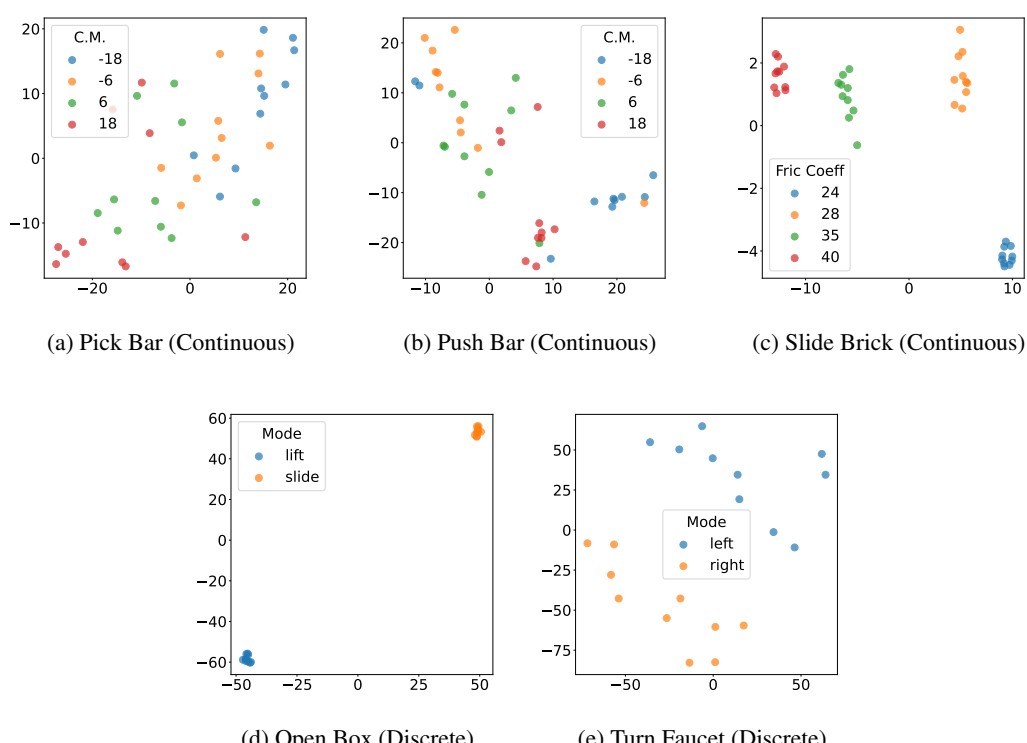

(a) Pick Bar (Continuous)  (b) Push Bar (Continuous)  (c) Slide Brick (Continuous)

(d) Open Box (Discrete)  (e) Turn Faucet (Discrete)

Figure 18: **Feature space visualization.** Top row: continuous parameter environments (Pick Bar, Push Bar, Slide Brick). Bottom row: discrete parameter environments (Open Box, Turn Faucet).

## D.2 TRAINING DATA AND SCALING EFFECTS.

The training data used in our main experiment is summarized in Table 6, where the "100%" row indicates the amount of data per environment. To account for the scarcity of data in real-world

Table 6: **Number of demonstrations used for each task.** Each cell shows failed/successful demonstrations.

|  | Push Bar | Pick Bar | Slide Brick | Open Box | Turn Faucet | Total |
|---|---|---|---|---|---|---|
| Main exp (100%) | 6000/250 | 6000/250 | 1560/130 | 20/20 | 20/20 | 13620/650 |
| 28% | 1000/250 | 1000/250 | 1300/130 | 20/20 | 20/20 | 3360/650 |

settings and to assess the effect of data scale on performance, we repeated the experiment with only 28% of the data. Results are presented in Table 7 and Figure 19.

Table 7: **Scaling effect on performance.** Reported metric is *Replans until Success* (↓, lower is better). We compare performance with 28% vs. 100% data across each environment.

(a) Push Bar

| Method | 28% | 100% |
|---|---|---|
| CQL | 12.69 | 9.96 |
| BC | 11.24 | 10.26 |
| AVDC | 5.65 | 6.83 |
| Ours | **3.91** | **4.26** |

(b) Pick Bar

| Method | 28% | 100% |
|---|---|---|
| CQL | 11.57 | 9.22 |
| BC | 8.88 | 8.79 |
| AVDC | 3.60 | 4.41 |
| Ours | **3.56** | **3.84** |

(c) Slide Brick

| Method | 28% | 100% |
|---|---|---|
| CQL | 12.33 | 8.94 |
| BC | **7.34** | 9.24 |
| AVDC | 10.75 | 7.36 |
| Ours | 11.42 | **6.69** |

(d) Open Box

| Method | 28% | 100% |
|---|---|---|
| CQL | 3.70 | 3.32 |
| BC | 3.55 | 1.59 |
| AVDC | 1.84 | 1.82 |
| Ours | **1.39** | **1.25** |

(e) Turn Faucet

| Method | 28% | 100% |
|---|---|---|
| CQL | 4.68 | 5.63 |
| BC | 4.15 | 3.54 |
| AVDC | 1.78 | 1.67 |
| Ours | **1.59** | **1.39** |

(f) Overall (Normalized)

| Method | 28% | 100% |
|---|---|---|
| CQL | 3.36 | 2.56 |
| BC | 2.58 | 1.98 |
| AVDC | 1.32 | 1.31 |
| Ours | **1.16** | **1.00** |

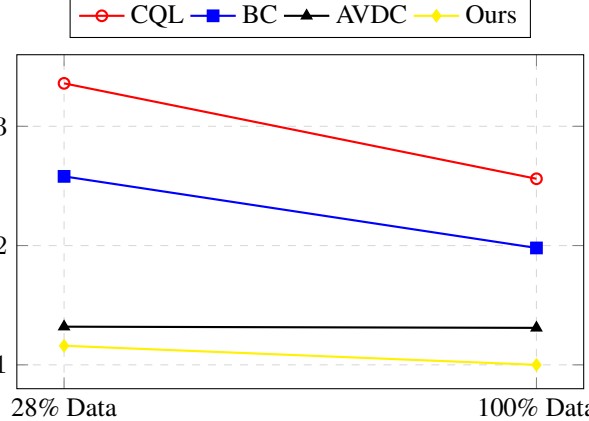

Figure 19: **Overall scaling effect.** Reported metric is *Replans until Success* (↓ lower is better).

With 28% of the data, our method surpasses baselines in 4 out of 5 tasks and achieves the best overall performance by a substantial margin, and the aggregate results show a clear advantage. As more data becomes available, our approach improves further, scaling smoothly and consistently across tasks. Taken together, these results demonstrate that our method delivers consistent and robust gains, both under limited data and when scaling up.

To furthur evaluate robustness under incomplete mode coverage, we include a harder triple-faucet task, where the robot must sequentially turn 3 faucets without knowing which side to push from for each of the faucets. This creates a task with **eight distinct task/system modes** and a longer planning horizon. We vary the size of the dataset and evaluate whether our retrieval process can still recover the correct configuration.

Table 8: **Data scaling effect on the 3-faucet task.** The table shows the exact match success rate between the hidden parameter of the current environment and that of the retrieved object embedding.

| Method \ Dataset Size (Successful, Failed) | (2, 14) | (10, 70) | (40, 280) | (120, 840) |
|---|---|---|---|---|
| Random | $11.87 \pm 1.28$ | $11.87 \pm 1.28$ | $11.87 \pm 1.28$ | $11.87 \pm 1.28$ |
| **Ours (Retrieval)** | $\mathbf{18.59 \pm 1.54}$ | $\mathbf{30.94 \pm 1.83}$ | $\mathbf{53.12 \pm 1.97}$ | $\mathbf{67.66 \pm 1.85}$ |

The results in Table 8 show that our method already provides meaningful improvements over the random baseline even with very small (not even fully covering the 8 modes of the task) datasets, and its performance scales steadily as more experience data becomes available. These findings indicate that our encoding method learns embeddings that remain robust in challenging, long-horizon environments, enabling the Retrieval Module to recover system parameters that closely match the true environment configuration.

### D.3 RUNTIME ANALYSIS

As shown in Table 9, we measured the average computation time over 100 iterations on a machine with an Intel Xeon W-2255 CPU and an NVIDIA RTX 3080 Ti GPU. Each iteration includes embedding retrieval, video generation (DDIM), plan selection (reject), and action decoding via optical flow. Although our full pipeline is slower per replan, this does not imply worse time-to-success. A more relevant metric for practical deployment is wall-clock time until the first successful execution, rather than latency per planning step. For example, in the Push Bar task, AVDC requires an average of 6.83 replans before success, with approximately 2.3 seconds per replan. Our full method requires fewer replans (4.26 on average), but each replan is slower (approximately 16.5 seconds).

Table 9: **Computation time (in seconds)** averaged over 100 trials.

| Component | AVDC | Ours w/o. Refine | Ours |
|---|---|---|---|
| Retrieve | N/A | $0.487 \pm 0.008$ | $11.876 \pm 0.036$ |
| DDIM | $0.777 \pm 0.001$ | $1.573 \pm 0.004$ | $1.599 \pm 0.006$ |
| Reject | N/A | $1.276 \pm 0.075$ | $1.230 \pm 0.068$ |
| Optical Flow | $1.564 \pm 0.021$ | $1.661 \pm 0.030$ | $1.761 \pm 0.028$ |
| Total | $2.341 \pm 0.021$ | $4.997 \pm 0.081$ | $16.466 \pm 0.082$ |

In addition, when a plan fails, execution requires approximately 25 seconds to terminate before replanning. Taking this into account, the expected time to first success is:

- AVDC: $6.83 \times (2.3 + 25) = 186.5$ seconds

- Ours: $4.26 \times (16.5 + 25) = 176.8$ seconds

Despite a higher per-replan cost, our method reaches the first success faster overall due to the substantially lower failure rate.

Moreover, our computational cost is amortizable, while execution time is not. With better hardware or a more accelerated algorithm, the refinement overhead can be reduced, whereas the real-world cost of a failed execution remains fixed. This means the time advantage of our method naturally increases with improved computing infrastructure.

Finally, our method provides an additional benefit that AVDC does not: once a state embedding is identified for a particular object, it can be reused in future interactions. This allows the agent to succeed with higher probability on subsequent trials without repeating the identification process, whereas AVDC would again incur the full failure cost of approximately 186.5 seconds to reach its next success.

## D.4 COMPARISON WITH LATENT-CONTEXT ADAPTATION METHODS.

Our problem setting is related to Hidden-Parameter MDPs and latent-context adaptation. We therefore include representative methods from this line of work, namely HiP-MDP (Killian et al., 2017) and PEARL (Rakelly et al., 2019), in our evaluation under the same setting as our main experiments.

We implement both methods as latent-context–conditioned actor–critic policies (HiP-MDP-AC and PEARL-AC) and evaluate them directly against our approach. Notably, these baselines rely on privileged supervision during training, including dense reward functions and action-level feedback, which are not available to our method. Beyond evaluating the original formulations, we also integrate the latent representations learned by these approaches into our framework by replacing our Retrieval Module with their inferred embeddings while keeping the remainder of the pipeline unchanged (HiP-MDP-ISE and PEARL-ISE). This setup examines whether prior latent-state estimation approaches can be directly incorporated into our system and benefit from our refinement and rejection mechanisms.

Table 10: Comparison with Latent-Context Adaptation Methods. **Metric:** Replans Until Success.

| Method | Push Bar | Pick Bar | Slide Brick | Open Box | Turn Faucet | Overall |
|---|---|---|---|---|---|---|
| HiP-MDP-AC | $8.65 \pm 0.24$ | $7.65 \pm 0.23$ | $11.29 \pm 0.22$ | $13.01 \pm 0.15$ | $13.03 \pm 0.14$ | $5.10 \pm 0.07$ |
| PEARL-AC | $8.64 \pm 0.23$ | $8.99 \pm 0.24$ | $13.20 \pm 0.13$ | $13.09 \pm 0.14$ | $2.57 \pm 0.09$ | $3.73 \pm 0.06$ |
| HiP-MDP-ISE | $7.88 \pm 0.24$ | $9.62 \pm 0.23$ | $10.35 \pm 0.23$ | $13.21 \pm 0.13$ | $13.06 \pm 0.14$ | $5.17 \pm 0.07$ |
| PEARL-ISE | $5.73 \pm 0.21$ | $5.70 \pm 0.21$ | $7.69 \pm 0.23$ | $1.60 \pm 0.04$ | $1.60 \pm 0.05$ | $1.28 \pm 0.04$ |
| **Ours** | $\mathbf{4.26 \pm 0.20}$ | $\mathbf{3.84 \pm 0.18}$ | $\mathbf{6.69 \pm 0.26}$ | $\mathbf{1.25 \pm 0.10}$ | $\mathbf{1.39 \pm 0.14}$ | $\mathbf{1.00 \pm 0.03}$ |

The results support two conclusions. First, our full video-planning system consistently outperforms latent-context adaptation baselines, even when those methods are given access to privileged supervision. Second, substituting external embeddings into our ISE framework improves performance over the original actor–critic implementations, indicating that our refinement and rejection procedures remain effective across different embedding representations.

Overall, these findings suggest that our performance gain is not attributable to a particular embedding choice, but to the integrated formulation that combines planning, retrieval, and adaptation. Our framework, therefore, empirically improves upon prior latent-state adaptation methods while remaining compatible with their learned representations.

## D.5 MULTI-FAUCET ENVIRONMENT AND ROBUSTNESS OF STATE EMBEDDING

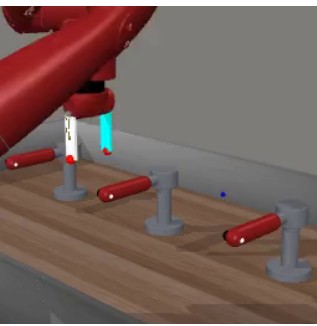

Figure 20: **Multi-faucet system identification setup.** A robot interacts with one of multiple faucet instances that share visual structure but differ in hidden physical properties (e.g., joint stiffness, friction, or rotation direction). The agent must infer the correct interaction mode through probing actions rather than relying on appearance alone.

To evaluate robustness in more complex and longer-horizon environments, we introduce an extended version of the faucet task in which the robot is required to turn three faucets sequentially. Similar to the original formulation, each faucet can only be rotated when pushed from the correct side; pushing from the wrong side causes the handle to jam and prevents further progress. This design induces

$2^3 = 8$ distinct interaction modes and significantly increases task difficulty due to both extended horizon and compounding failure modes.

We use this environment to examine whether the proposed state embedding encoding method can still recover the correct hidden system parameters under more challenging conditions. State embeddings are constructed using datasets of varying sizes, after which we apply the same Retrieval procedure as described in Section 4.1 using newly collected interaction trials. We then compare the retrieved embedding with the ground-truth hidden configuration of the environment and report the success rate of exact matches.

Table 11: **State Embedding Robustness in Multi-Faucet Environment.** The hidden parameter matching success rate over 640 trials.

| Method / Dataset Size (Success, Fail) | (2, 14) | (10, 70) | (40, 280) | (120, 840) |
|---|---|---|---|---|
| Random | $11.87 \pm 1.28$ | $11.87 \pm 1.28$ | $11.87 \pm 1.28$ | $11.87 \pm 1.28$ |
| **Ours (Retrieval)** | $\mathbf{18.59 \pm 1.54}$ | $\mathbf{30.94 \pm 1.83}$ | $\mathbf{53.12 \pm 1.97}$ | $\mathbf{67.66 \pm 1.85}$ |

The results demonstrate that the proposed encoding method already yields meaningful improvements over random selection with very limited data and continues to improve steadily as more experience becomes available. This indicates that the learned embeddings remain informative under long-horizon dynamics and combinatorial interaction modes, enabling the Retrieval Module to recover system parameters that closely match the underlying environment configuration.

### D.6 ALTERNATIVE PLAN SELECTION STRATEGY

We consider an alternative plan selection strategy that uses the best retrieved successful video as a positive signal to guide planning. In this variant, the video retrieved by the Retrieval Module is treated as an attractor and directly influences plan selection by favoring candidates that are closer to this favorable outcome. We refer to this strategy as *Attract*.

We compare this approach against our default Reject strategy, which explicitly filters out failed candidates through rejection during sampling. Both strategies share the same planning architecture and state embedding mechanism; the only difference lies in how retrieved information is used during selection. While Attract biases sampling toward favorable outcomes, Reject instead eliminates implausible or inconsistent candidates. The methods are evaluated over 80 trials for each task.

Table 12: **Attract vs. Reject strategies.**

| Method | Push Bar | Pick Bar | Slide Brick | Open Box | Turn Faucet | Overall (Normalized) |
|---|---|---|---|---|---|---|
| Attract | $5.55 \pm 0.55$ | $4.16 \pm 0.45$ | $12.26 \pm 0.52$ | $1.31 \pm 0.26$ | $1.31 \pm 0.29$ | $1.07$ |
| **Reject (Ours)** | $\mathbf{5.95 \pm 0.55}$ | $\mathbf{4.35 \pm 0.47}$ | $\mathbf{10.60 \pm 0.58}$ | $\mathbf{1.13 \pm 0.25}$ | $\mathbf{1.14 \pm 0.27}$ | $\mathbf{1.00}$ |

The results in Table 12 show that the rejection-based strategy consistently outperforms attraction-based selection. Although Attract performs competitively on several tasks, it exhibits significantly worse performance on Slide Brick, the most challenging task in our evaluation suite. This indicates that a purely attraction-based signal may be insufficient under difficult dynamics where distinguishing incorrect plans is more critical than favoring a single successful example.

Note that rejection and attraction are not mutually exclusive. Rejection provides a repulsive signal that removes incompatible hypotheses, while attraction offers a directional signal toward plausible regions of the solution space. A combined approach that integrates both signals may further improve sampling efficiency and accelerate adaptation, which we leave as an interesting direction for future work.

### D.7 ROBUSTNESS TO TASK-IRRELEVANT VISUAL DISTRACTORS

To evaluate robustness under visually complex conditions, we introduce two noisy variants of the Slide Brick and Open Box environments by randomly altering background appearances to simulate task-irrelevant visual signals. Such distractors are common in real-world settings and may interfere

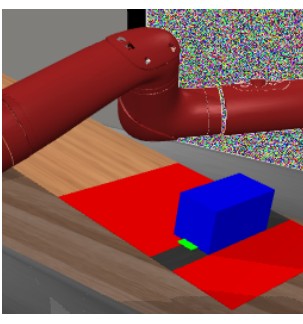 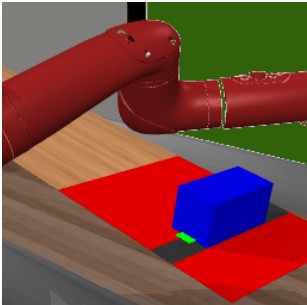

(a) Pixel-level noise           (b) Uniform noise

Figure 21: **Robustness to task-irrelevant visual noise.** We introduce appearance-randomized environments with visually distracting backgrounds while preserving identical interaction dynamics. This setting isolates whether distance functions respond to task-relevant motion rather than background variation.

with similarity metrics that rely on raw pixel comparisons. Table 13 reports the number of replans required before success under these noisy conditions.

Table 13: **Performance under visually noisy environments.** Metric: Replans until success.

| Method | Slide Brick Noisy | Open Box Noisy |
|---|---|---|
| No Rejection | 6.09 | 1.91 |
| DINO | 5.56 | 1.65 |
| L2 error | 4.89 | 1.79 |
| Seg + DINO | 5.54 | **1.65** |
| Seg + L2 error | **4.53** | **1.65** |

The results show that while L2 distance underperforms DINO on Open Box, it performs better on Slide Brick. A plausible explanation is that DINO embeddings prioritize semantic abstraction and may discard fine-grained spatial information that is critical for estimating hidden parameters in tasks that require high geometric precision. In contrast, pixel-level L2 distance preserves low-level structural cues that remain informative even under moderate visual noise.

Furthermore, we demonstrate a simple and practical strategy for mitigating task-irrelevant signals by introducing segmentation to isolate task-relevant regions. By masking out background distractors, the distance metric, whether based on L2 or DINO features, can focus on the portions of the image that encode the underlying system parameters. This approach is feasible in practice, as high-quality segmentation masks can be obtained using modern vision-language models such as Segment Anything Models (SAMs). These results suggest that segmentation provides an effective mechanism for improving robustness in visually cluttered environments without requiring changes to the overall planning framework.

### D.8 GENERALIZATION OF STATE EMBEDDINGS TO OBJECTS VARIANCE

Our approach improves adaptation by retrieving state embeddings from previously observed objects and using them to approximate the hidden system parameters of the environment. When encountering a new object at test time, interaction mechanisms that were learned before (for example, push vs. pull for doors, or slide vs. lift for lids) can be recalled and transferred to guide behavior.

To evaluate generalization, we construct novel environments in which object appearances differ from training, while the underlying interaction mechanisms remain unchanged. Each stored state embedding is associated with its ground-truth system parameters for evaluation. During testing, we retrieve the state embedding using the same retrieval procedure as in the main experiment and compare its associated parameters with those of the current environment. We consider the following baselines:

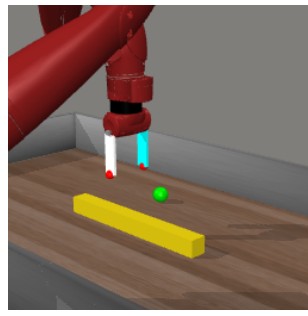

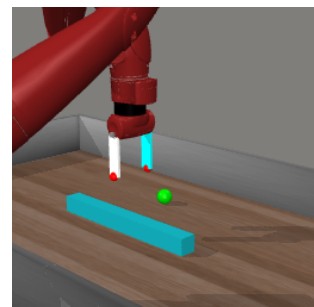

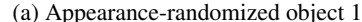

(a) Appearance-randomized object 1         (b) Appearance-randomized object 2

Figure 22: **Appearance variation under fixed interaction dynamics.** Object color and texture are randomized while geometry and physical properties remain unchanged. This setting decouples visual appearance from interaction behavior and tests adaptation under perceptual aliasing.

**Random Guess:** Randomly select one embedding from the database.

**ISE-Seen:** Perform retrieval using embeddings constructed from environments seen during training.

**ISE-Unseen:** Perform retrieval on novel environments with unseen object appearances.

Table 14: Raw retrieval error on hidden system parameters (lower is better).

| Method | Push Bar | Open Box | Pick Bar | Turn Faucet | Slide Brick |
|---|---|---|---|---|---|
| Random Guess | $.0954 \pm .0052$ | $.4900 \pm .0500$ | $.0954 \pm .0052$ | $.4900 \pm .0500$ | $.0528 \pm .0040$ |
| ISE-Seen | $.0694 \pm .0057$ | $.0200 \pm .0140$ | $.0793 \pm .0069$ | $.0200 \pm .0140$ | $.0448 \pm .0032$ |
| ISE-Unseen | $.0693 \pm .0049$ | $.0700 \pm .0255$ | $.0729 \pm .0054$ | $.0200 \pm .0140$ | $.0428 \pm .0032$ |

Table 15: **Normalized Performance.** Random Guess is normalized to 0, and ISE-Seen is normalized to 1.

| Method | Push Bar | Open Box | Pick Bar | Turn Faucet | Slide Brick | Oevrall |
|---|---|---|---|---|---|---|
| Random Guess | 0.0000 | 0.0000 | 0.0000 | 0.0000 | 0.0000 | 0.0000 |
| ISE-Seen | 1.0000 | 1.0000 | 1.0000 | 1.0000 | 1.0000 | 1.0000 |
| ISE-Unseen | 1.0028 | 0.8936 | 1.3971 | 1.0000 | 1.2468 | 1.1081 |

# E   THEORETICAL JUSTIFICATION

This section provides a simplified probabilistic analysis that explains why the rejection mechanism improves the likelihood of selecting the correct interaction hypothesis.

## E.1   PRELIMINARIES

Let
$$\mathcal{K} = \{1, \ldots, K\}$$
denote a finite set of latent interaction modes. The planner maintains a categorical belief distribution
$$q = (q_1, \ldots, q_K), \quad q_k = \Pr(\text{mode } k).$$
Let the true interaction mode be $k^\star$ with probability
$$q_{k^\star} \in (0, 1).$$

At each replanning step, the planner draws $N$ independent hypotheses

$$K_1, \ldots, K_N \sim q,$$

generates predicted outcomes (rollouts) for each sampled mode, and selects the hypothesis whose rollout is maximally dissimilar from the previously failed execution.

### E.2 REJECTION AS PROBABILITY AMPLIFICATION

We assume a mild separation condition: rollouts generated from the same failing mode remain closer to the failed execution than rollouts generated from different modes. Under this assumption, rejection selects the correct mode whenever $k^\star$ appears at least once among the $N$ sampled hypotheses.

The probability that at least one of the $N$ samples equals $k^\star$ is

$$p_{\mathrm{rej}} = 1 - (1 - q_{k^\star})^N.$$

For any $N \geq 2$, we have $p_{\mathrm{rej}} > q_{k^\star}$ whenever $q_{k^\star} \in (0, 1)$, showing that rejection strictly amplifies the probability of selecting the correct interaction mode.

### E.3 CONVERGENCE UNDER REPLANNING

Each replanning attempt succeeds with probability at least $p_{\mathrm{rej}}$. Assuming independence across replanning attempts, the probability that no success is observed after $M$ attempts is

$$\Pr(\text{no success after } M) = (1 - p_{\mathrm{rej}})^M.$$

Using the standard tangent-line bound

$$1 - x \leq e^{-x} \quad \text{for all } x \in [0, 1],$$

we obtain the exponential tail bound

$$\Pr(\text{no success after } M) \leq e^{-p_{\mathrm{rej}} M}.$$

Thus, the probability of failure decays exponentially in the number of replanning attempts $M$, with rate parameter $p_{\mathrm{rej}}$. Moreover, $p_{\mathrm{rej}}$ is a monotonically increasing function of the number of sampled hypotheses $N$:

$$\frac{\partial}{\partial N} \big[ 1 - (1 - q_{k^\star})^N \big] = (1 - q_{k^\star})^N \log\big( \tfrac{1}{1 - q_{k^\star}} \big) > 0,$$

so drawing more hypotheses per replanning step strengthens the exponential bound above.

### E.4 DISCUSSION

This analysis shows that the proposed rejection strategy does not merely resample hypotheses, but actively amplifies the probability of selecting the correct interaction mode by eliminating inconsistent candidates. Although the argument relies on an idealized separation assumption, it provides a theoretical explanation for the empirical reduction in the number of replans observed in ambiguous and long-horizon environments.

## F LIMITATION

To isolate the challenge of video-based belief refinement, our work assumes a reliable action module, attributing failures solely to planning errors. While this abstraction simplifies evaluation, it omits real-world execution noise, which future work could address through uncertainty-aware control. Our visually driven approach enables interpretability but struggles with visually ambiguous tasks

(e.g., subtle directional differences); incorporating tactile or proprioceptive sensing could improve disambiguation. We also observe a "first-frame bias" in the video generator under low-data regimes, where plans become overly deterministic. Although we have shown in Section 5.6 that such bias can be mitigated via increasing the number of candidate plans $n$ and noise injection to the first-frame image, future methods may benefit from diversity-promoting strategies or explicit mode learning.

## G    USE OF LLMS

In accordance with ICLR policy on the use of large language models (LLMs), we disclose that LLMs were used only for minor text polishing to improve grammar and readability. They were not involved in research ideation, experiment design, analysis, or the generation of scientific content. All scientific contributions are solely those of the authors, who take full responsibility for the contents of this paper.

