# OpenReview forum: "Implicit State Estimation via Video Replanning"
_ICLR.cc/2026/Conference — Submitted to ICLR 2026_

### Official Review · Reviewer_1Eqz · 2025-10-21

**Soundness:** 3
**Presentation:** 1
**Contribution:** 3
**Rating:** 6
**Confidence:** 4

**Summary:**

The paper proposes a novel method for trajectory planning via video generation. The idea is to use a diffusion video generation model to generate a video of successful task completion given an image of a state and a goal. In difference to prior methods, the key innovation of this paper is to introduce a replanning stage that updates a state latent based on prior experiences and an offline data buffer. The results show that replanning steps can be reduced by using these buffers, as they enable to select those generated trajectories that are unlikely to fail. The evaluation compares to some BC and RL baselines as well as a prior video prediction method which seems to have inspired this work.

**Strengths:**

- I like the brick sliding task. It seems like its a bit more challenging than the other tasks. I wonder how the demonstrations shape the performance on this task.
- The evaluation is sufficient in terms of number of baselines and tasks I think. On top, the work performs real-world experiments which is always nice to see.
- The introduction is well written.

**Weaknesses:**

# Method
- The method uses video as an action representation, similar to other recent works that use pixel-based representations. However, in the presented form those are limited to quasi-static manipulations if I look at the experiments (apart from brick sliding maybe?). I would mention that this representation is not without limitations despite its merits.
- I think claiming that you introduce a novel benchmark is a slight overstatement, because you essentially seem to modify some tasks from metaworld. You can just say you evaluate your method.

# Wording:
- The writing is pretty sloppy in places. I think individual typos are not an issue which is why I listed them under minor, but in their sum, there are quite a lot of them. Especially the related work section has quite a few typos and mistakes that must be fixed.

# Evaluation
- How do you collect the offline interactions? How robust is the system to the choice of demos here?
- It seems like the method gets access to an offline dataset, which could be seen as access to additional trials. How would the performance compare to the baselines if you only provide access to the actual previously failed interactions until M=14?
- I am not sure how meaningful the results are in Table 3. Shouldnt the plans be semantically similar to the GT plans and not visually? Especially SSIM can be quite prone to some minor details here. What about clip embedding similarities or something like that? I like the general idea of the experiment, but I am not sure if visual similarity is the right metric.
- I think having some other baselines that test video retrieval methods might be interesting, but it is not a big weakness.

# Clarity
- You distinguish between plans and interactions without rigorously defining the two concepts. To me it is not completely clear what sets them apart. Can you explain this further? This would improve the clarity for the reader in my eyes.
- In the abstract and title you state that your method is "implicit" as it does not directly model the unknown state variables. Yet you state in Section 4 that you encode "a hypothesis of the hidden parameters" and that you "estimate[...] hidden parameters from failed interactions". I think I follow you in saying that your method does not model hidden parameters in a way as Memmel et al (2024) for instance, but then I would reformulate the above sentences in Section 4 to improve clarity on this.
- I am not 100% sure if I understand the refinement procedure of the method "Ours (Refine)". What do you mean by refining from scratch? This part of the paper should be clarified slightly in wording I think.
- The related work is a bit off in my eyes. Even if you compare to offline RL, I think your method really does not fall into that category. Instead I think you might be better off at discussing other recent works that use video retrieval for few-shot adaptation, eg [1, 2].
- Table 5 us difficult to understand and should be improved. The same holds for Table 6.

# Minor
- "people have explored using video to train dynamics model" in line ~84. --> should be plural: "models", same for reward models.
- " to guide downstream policy" in line 88 --> " to guide a downstream policy"
- " Fisher Information Matrix" in line ~92 should be " Fisher information matrix"
- " The paradigm of offline RL has existed for dacades" in line ~111 should be "decades".
- "In addition, with the development of large language model and transformer, there are also some research about using transformer for reinforcement learning" in line 114. This sentence needs reformulation. Should be LLMs and transformers in plural and there "is" research.
- "our framework do not assume" in line 116 --> "does not assume"
- "RL methods" in line ~76 --> introduce abbreviation even if its clear in this context.
- "We consider a setting where certain system parameters θ, such as mass, friction, or utility" in line ~120. Utility is not a system but rather a task parameter I would say.
- " We complement our focus on plan evaluation by using a fixed action prediction module to convert videos into actions for execution." I am not sure if "complement" is the right word here?
- " you often don’t know its exact weight" --> " you often do not know its exact weight"
- Sometimes you italicize the dataset variable D and sometimes you use \mathcal{D}. Be consistent here.
- " where the Policy takes in state and state embedding directly" in line 246. Dont capitalize policy here
- Line 422 has a double dot at the end.
- Line 391 "Aciton" -> "Action"

---
# Sources
[1] Haldar, Siddhant, et al. "Teach a Robot to FISH: Versatile Imitation from One Minute of Demonstrations." _Robotics: Science and Systems_. 2023.
[2] Memmel, Marius, et al. "STRAP: Robot Sub-Trajectory Retrieval for Augmented Policy Learning." _The Thirteenth International Conference on Learning Representations_. 2024.

**Questions:**

- Could the method also work using only its own failed attempts and no extra offline dataset?
- How would your selection mechanism compare to selecting the action thats closest to the best succeding demo instead of picking the one furthest away from failure?
- Does you method suffer from compounding errors under long action sequences? How long can you make those sequences?

---

> ### Author Response · Authors · 2025-11-28
> **Response to reviewer 1Eqz (1/3)**
>
> We appreciate the reviewer's detailed and constructive feedback. We address each of the reviewer's concerns below.
>
> ### Video as action representation
> > The method uses video as an action representation, similar to other recent works that use pixel-based representations. However, in the presented form, those are limited to quasi-static manipulations if I look at the experiments (apart from brick sliding maybe?). I would mention that this representation is not without limitations despite its merits.
>
> We agree with the reviewer that using video as an action representation has its limitations. Incorporating additional modalities for both the state estimator and the low-level policy (the Action Module) may enable the system to handle more challenging tasks, which is also what we are actively investigating in follow-up work.
>
> ### Toning Down The Meta-World Benchmark
>
> We thank the reviewer for this feedback and have revised the paper to tone down the claim accordingly. Specifically, we now state that we introduce a new task set called "Meta-World System Identification Suite" within the Meta-World environment for evaluating adaptation, rather than presenting it as an entirely new benchmark.
>
> ### Wording/Typos
>
> We thank the reviewer for carefully pointing out these issues. We have corrected all listed typos, standardized notation, and revised the related work section for clarity.
>
> ### Data Collection
>
> > *How do you collect the offline interactions? How robust is the system to the choice of demos here?*
>
> In simulation, we use scripted policies to collect offline interaction data. Specifically, the policy takes as input both the current observation and a “guessed” system parameter. For example, in the Bar Picking task, the guess is a one-dimensional scalar representing the position of the center of mass relative to the bar. If this guess matches the true underlying parameter of the environment (e.g., both are centered), the interaction results in a successful video; otherwise, it produces a failed interaction. These outcomes correspond directly to the success signal $s_i$ described in Section 3.
>
> For each object, we perform multiple interactions with different guesses, generating several failed trials, followed by an interaction using the correct guess, which produces a successful trial. We then assign all trials from the same object the same object ID $o_i$ and terminate the interaction loop for this object. The policy then starts interacting with a new object by randomizing the initial configuration and hidden parameters. As for real-world experiments, we simply used tele-operation to replace the guess-augmented scripted policy.
>
> Regarding robustness to the choice of demonstrations, our method only assumes that each task/system mode is reasonably covered by the offline dataset. As long as a task/system mode has been observed, the retrieval module can identify and reuse the corresponding state embedding at test time. For example, in the Bar Pushing task, as long as the dataset contains interactions spanning different center-of-mass configurations, the system can recover the correct task/system mode for a novel object through retrieval.
>
> To evaluate robustness under *incomplete mode coverage*, we include a harder triple-faucet task with **eight distinct task/system modes** and a longer planning horizon. We vary the size of the dataset and evaluate whether retrieval can still recover the correct configuration.
>
> **Metric: Hidden Parameter Matching Success Rate (%)**
> Evaluated over 640 trials.
>
> | Method \ Dataset Size (Successful, Failed) | (2, 14) | (10, 70) | (40, 280) | (120, 840) |
> |--------------------------------------------------------|---------|----------|-----------|------------|
> | Random                                                 | 11.87 ± 1.28 | 11.87 ± 1.28 | 11.87 ± 1.28 | 11.87 ± 1.28 |
> | **Ours (Retrieval)**                                   | **18.59 ± 1.54** | **30.94 ± 1.83** | **53.12 ± 1.97** | **67.66 ± 1.85** |
>
> In the smallest dataset, only **2 out of 8 task/system modes** are observed. Even in this case, retrieval already improves hidden-parameter recovery over baseline and continues to scale reliably as more interaction data becomes available. This shows that the system does not require exhaustive coverage to be effective and degrades gracefully when data is limited.
>
> Finally, when retrieval is imperfect or unavailable, the rejection module continues to operate in a training-free manner by filtering implausible plans based on visual consistency rather than learned retrieval alone. This allows the system to remain functional even when the dataset coverage is incomplete. Please refer to Appendix D.5 for more details about this experiment.
>
> We thank the reviewer for raising this insightful question. We have included the discussion regarding the data collection process and the robustness study in the appendix section.

---

> ### Author Response · Authors · 2025-11-28
> **Response to reviewer 1Eqz (2/3)**
>
> ### Video metrics
> We agree with the reviewer that semantic similarity may be a more appropriate metric than pure visual similarity in this context. We have therefore added CLIP-score and DINO-score (cosine similarity between CLIP and DINOv2 embeddings) to the evaluation. With these additional metrics, DINOv2 still produces the best results, which supports our original conclusion.
>
> | Method \ Metric        | PSNR ↑ | SSIM ↑ | LPIPS ↓ | *CLIP-score↑* | *DINO-score↑* |
> |----------------|--------|--------|---------| --- | --- |
> | AVDC           | 19.426 | 0.700  | 0.271   | 0.8697 | 0.7051
> | Ours (R3M)     | 19.632 | 0.704  | 0.261   | 0.8745 | 0.7049 |
> | Ours (CLIP)    | 19.584 | 0.703  | 0.262   | 0.8712 | 0.7068 |
> | **Ours (DINOv2)** | **19.817** | **0.708** | **0.255** | **0.8822** | **0.7168** |
>
>
> ### Clarification
>
> We thank the reviewer for pointing out the ambiguities and appreciate their constructive advice. We have made changes accordingly:
>
> ***Difference between plans/interactions***
> > You distinguish between plans and interactions without rigorously defining the two concepts. To me it is not completely clear what sets them apart. Can you explain this further? This would improve the clarity for the reader in my eyes.
>
> We distinguish between plans and interactions because they originate from fundamentally different sources (the **generative model** vs. the **environment**) and therefore play different roles in our framework, even though both are represented in the same video modality.
>
> Specifically, a **plan** is produced by the **generative model**. Because the planner is trained only on successful trajectories, it tends to generate videos that resemble successful executions. However, when its internal representation of task or system parameters is inaccurate, it may *hallucinate* and generate implausible plans. For example, it may generate a video of pushing a door open when the door should instead be pulled. In such cases, disagreement between the plan and actual execution can be observed (e.g., the door remains stuck during execution).
>
> In contrast, an **interaction** is a real execution observed directly from the **environment**. As a result, interactions always reflect the true underlying task or system configuration. While interactions may be noisy or partially ambiguous, they are never physically incorrect in the way generated plans can be.
>
> ***Why our approach is“implicit”***
> > In the abstract and title you state that your method is "implicit" as it does not directly model the unknown state variables. Yet you state in Section 4 that you encode "a hypothesis of the hidden parameters" and that you "estimate[...] hidden parameters from failed interactions". I think I follow you in saying that your method does not model hidden parameters in a way as Memmel et al (2024) for instance, but then I would reformulate the above sentences in Section 4 to improve clarity on this.
>
> We have revised section 4 to avoid the language use that would connect to explicit state estimation to improve clarity.
>
> ***The “Refinement” process***
> > I am not 100% sure if I understand the refinement procedure of the method "Ours (Refine)". What do you mean by refining from scratch? This part of the paper should be clarified slightly in wording I think.
>
> By “refining from scratch,” we mean that the refinement procedure is initialized without a retrieved state embedding.
>
> Concretely, we model the embedding space using a **multivariate Gaussian prior** with per-dimension variance estimated from the dataset. During pre-processing, we compute the empirical mean and per-dimension variance of all state embeddings, and use this distribution as a prior over valid system parameters. At test time, if no retrieved embedding is available (or the retrieval confidence is low), we sample an initial embedding from this prior and use it to initialize the refinement process.
>
> We have revised the paper to clarify that refinement always operates in the same embedding space; the only difference is whether initialization comes from retrieval or from the learned prior.
>
> ***Related works***
> > The related work is a bit off in my eyes. Even if you compare to offline RL, I think your method really does not fall into that category. Instead I think you might be better off at discussing other recent works that use video retrieval for few-shot adaptation, eg [1, 2].
>
> We thank the reviewer for the suggestion. We polished our related works and included a new section to talk about recent video-based few-shot adaptation methods and prior latent-context adaptation approaches.
>
>
>
>
> ***Tables 5 and 6***
> > Table 5 us difficult to understand and should be improved. The same holds for Table 6.
>
> We thank the reviewer for pointing this out. We have revised the tables 5 and 6 to improve clarity.

---

> ### Author Response · Authors · 2025-11-28
> **Response to reviewer 1Eqz (3/3)**
>
> ### Other test-time retrieval baselines
>
> We thank the reviewer for suggesting additional retrieval baselines. To address this, we implemented two prior latent-context estimation methods [1,2] as test-time estimators in place of our retrieval process, while keeping the rest of the pipeline (refinement and rejection) unchanged. The results show that our method still outperforms these alternatives, confirming the effectiveness of our approach. Please refer to Appendix D.4 for more details about this experiment.
>
> | Method        | Push Bar       | Pick Bar       | Slide Brick     | Open Box       | Turn Faucet     | Overall
> |---|---|---|---|---|---|---
> | HiP-MDP-ISE  | 7.88 ± 0.24    | 9.62 ± 0.23    | 10.35 ± 0.23    | 13.21 ± 0.13   | 13.06 ± 0.14    | 5.17 ± 0.07
> | PEARL-ISE    | 5.73 ± 0.21    | 5.70 ± 0.21    | 7.69 ± 0.23    | 1.60 ± 0.04    | 1.60 ± 0.05     | 1.28 ± 0.04
> | **Ours**     | **4.26 ± 0.20** | **3.84 ± 0.18** | **6.69 ± 0.26** | **1.25 ± 0.10** | **1.39 ± 0.14** | **1.00 ± 0.03**
>
> ### Performance w/o the offline dataset
>
> We thank the reviewer for raising this question. If the offline dataset is no longer available at test time, the retrieval process could no longer function. In such cases, our method will fall back to only refinement+rejection. As clarified above, the refinement process can still function without a retrieved state embedding by treating the embedding space of all state embeddings as a multivariate Gaussian prior and sampling from it. Such a baseline is already included in the main experiment, dubbed Refine (Scratch).
>
> ### Selection mechanism
> > How would your selection mechanism compare to selecting the action that's closest to the best succeeding demo instead of picking the one furthest away from failure?
>
> We thank the reviewer for suggesting this insightful comparison. During selection, one could use the best succeeding video retrieved by the Retrieval Module and use it as a positive signal to guide plan selection. Motivated by this suggestion, we consider an alternative plan selection strategy that uses the successfully retrieved video from the Retrieval Module as a positive signal during planning. We refer to this variant as Attract. We then compare it with our failed-plan rejection strategy, denoted as Reject, in the table below. The Attract baseline follows our full framework, but replaces the Rejection Module with this attraction-based selection mechanism.
>
> | Method                       | Push Bar    | Pick Bar    | Slide Brick  | Open Box    | Turn Faucet | Overall (Normalized) |
> | ---------------------------- | ----------- | ----------- | ------------ | ----------- | ----------- | -------------------- |
> | Attract           | 5.55 ± 0.55 | 4.16 ± 0.45 | 12.26 ± 0.52 | 1.31 ± 0.26 | 1.31 ± 0.29 | 1.07      |
> | **Reject (Ours)** | 5.95 ± 0.55 | 4.35 ± 0.47 | 10.60 ± 0.58 | 1.13 ± 0.25 | 1.14 ± 0.27 | **1.00**             |
>
> Overall, our rejection-based approach outperforms the attraction strategy. Although the attraction method performs competitively on several tasks, it falls notably short on **Slide Brick**, which is the most challenging task in our suite. This highlights the robustness of the rejection strategy, especially in harder environments.
>
> Finally, we emphasize that the two strategies are fully **composable**. The rejection strategy provides a repulsive signal in the sampling space, while the attraction strategy offers a plausible goal to pull toward. Finding a way to effectively combine and balance both signals could further improve sampling efficiency and accelerate adaptation, which we see as a promising direction for future work.
>
> ### Compounding action errors
>
> While predicting long action sequences in an open-loop manner can indeed suffer from compounding errors, our method, like most video-planning approaches, operates in a closed-loop fashion. The video generator first produces a video plan that **spans the entire episode.** This plan is then converted into a sequence of target point trajectories, which are tracked by a low-level policy rather than executed open-loop.
>
> At each timestep, the Action Module receives fresh observations and predicts the next action based on how closely the current state follows the planned trajectory. Because execution is continuously corrected online using feedback from the environment, actions are not rolled out as a fixed sequence. As a result, compounding action errors are largely mitigated.
>
>
> ### References
>
> [1] Killian, Taylor W. et al. “Robust and Efficient Transfer Learning with Hidden Parameter Markov Decision Processes.” Advances in neural information processing systems 30 (2017): 6250-6261 .
>
> [2] Rakelly, Kate et al. “Efficient Off-Policy Meta-Reinforcement Learning via Probabilistic Context Variables.” International Conference on Machine Learning (2019).

---

### Official Review · Reviewer_DZm1 · 2025-10-23

**Soundness:** 2
**Presentation:** 2
**Contribution:** 2
**Rating:** 6
**Confidence:** 2

**Summary:**

This work introduces a novel video planning framework that enables robots to adapt to uncertain environments by learning from past failures. Instead of explicit system identification, the proposed method performs implicit state estimation by integrating interaction-time data to update model parameters and filter out unsuccessful strategies.

The proposed system uses a retrieval mechanism to refine its understanding of hidden parameters (e.g., mass, friction) and a rejection strategy to avoid repeating failed plans. This allows for rapid online adaptation without prior knowledge of the environment's dynamics.

The proposed framework has been validated only on two scenarios: a simulated benchmark and a real-world door-opening task. The results show that in the two considered scenarios, it significantly reduces replanning attempts and outperforms state-of-the-art video planning and reinforcement learning baselines.

**Strengths:**

The paper addresses the interesting problem of the inability to adapt to failures and reason about uncertainty during interaction. It provides a framework that leverages past interaction videos and a simple rejection mechanism to guide future plans, mimicking human trial-and-error, while other approaches use complex explicit belief models.

The paper provides a thorough experimental evaluation, but only in two scenarios: their new simulation benchmark (Meta-World System Identification) and the real problem of opening a closed door. The experiments show clear quantitative improvements over strong baselines (AVDC, BC, CQL) in these two scenarios in reducing replans.

**Weaknesses:**

The framework attributes all failures to planning errors, assuming a near-perfect action module that can always execute the generated video plan. This omits real-world execution noise and control uncertainties, which could cause failures regardless of plan quality.

The method relies solely on vision. It may struggle with tasks where the crucial state parameter is visually ambiguous (e.g., two objects with identical appearance but different masses) or requires other sensory modalities like touch or force-feedback.

While still practical, the full method (with retrieval and generating multiple candidate plans) is slower than the baseline AVDC (~5 sec vs ~2.3 sec per replan). The refinement process adds significant latency (~16.5 sec), making it less suitable for real-time tasks.

The video generator can produce overly deterministic plans conditioned heavily on the initial observation, especially with limited data. While mitigated with noise injection, it remains a limitation that needs further analysis.

The retrieval-based state estimation requires storing and accessing the entire experience dataset at test time, which may not be scalable or efficient for long-term deployment compared to a fully learned belief state. No proper discussion on this problem has been conducted and analysed.

Last, the notion of planning used in the paper is not the classical one used, for instance, in the AI Planning community, and it seems more related to motion planning. The authors shall better characterize their notion of planning, and the relation that there may exist with POMDP planning (i.e. synthesize a policy to achieve a goal in a partially observable Markov decision problem).

The additional material, consisting of just a link to another web page, is not informative enough to enable an individual to reproduce the paper's results.

**Questions:**

The questions are to comment on the 7 weak points identified in the previous section.

---

> ### Author Response · Authors · 2025-11-28
> **Response to reviewer DZm1 (1/2)**
>
> We thank the reviewer for raising their concerns. Below, we address each question in detail.
>
> ### Assumption of good action module
> We agree with the reviewer that a poor Action Module can lead to failures regardless of the quality of the plan. We would like to restate that the core focus of this work is to improve the quality of video plans, and the failure mode described above is outside the scope of this paper. However, we agree that improving the Action Module is an important direction for future work, and we are in fact actively investigating this direction in follow-up work.
>
>
> ### Vision-only approach
> > The method relies solely on vision. It may struggle with tasks where the crucial state parameter is visually ambiguous (e.g., two objects with identical appearance but different masses) or requires other sensory modalities like touch or force-feedback
>
> We would like to clarify that our task suite does contain tasks with identical object appearances but different system parameters. In such cases, the system parameters cannot be inferred from static images, and this is precisely why we introduce the use of past interaction videos for identifying them. For example, in the *Slide Brick* task, the planner must reason about the friction between the slope and the object, which is not directly observable from images but can be identified through video. That said, we agree with the reviewer that some system parameters may be difficult to infer from video alone, and incorporating additional modalities would be a promising direction for future work. Importantly, the core insights of our framework, including model-free retrieval, model-based refinement, and failed-plan rejection, are not tied to the video modality and can naturally extend to these additional modalities.
>
> ### Computational overhead
> > While still practical, the full method (with retrieval and generating multiple candidate plans) is slower than the baseline AVDC (5 sec vs ~2.3 sec per replan). The refinement process adds significant latency (16.5 sec), making it less suitable for real-time tasks.
>
> We would like to emphasize that although our full pipeline is slower per replan, this does not imply worse time-to-success. A more relevant metric for practical deployment is wall-clock time until the first successful execution, rather than latency per planning step. For example, in the Push Bar task, AVDC requires an average of 6.83 replans before success, with approximately 2.3 seconds per replan. Our full method requires fewer replans (4.26 on average), but each replan is slower (approximately 16.5 seconds).
>
> In addition, when a plan fails, execution requires approximately 25 seconds to terminate before replanning. Taking this into account, the expected time to first success is:
> - AVDC: 6.83 × (2.3 + 25) = 186.5 seconds
> - Ours: 4.26 × (16.5 + 25) = 176.8 seconds
>
> Despite higher per-replan cost, our method reaches the first success faster overall due to the substantially lower failure rate.
>
> Moreover, our computational cost is amortizable, while execution time is not. With better hardware or more accelerated algorithm, the refinement overhead can be reduced, whereas the real-world cost of a failed execution remains fixed. This means the time advantage of our method naturally increases with improved compute infrastructure.
>
> Finally, our method provides an additional benefit that AVDC does not: once a state embedding is identified for a particular object, it can be reused in future interactions. This allows the agent to succeed with higher probability on subsequent trials without repeating the identification process, whereas AVDC would again incur the full failure cost of approximately 186.5 seconds to reach its next success.
>
> Therefore, we respectfully disagree that our method is slower than the baselines in real-time executions. We have included this discussion in the run-time analysis section.
>
> ### First frame bias
> > The video generator can produce overly deterministic plans conditioned heavily on the initial observation, especially with limited data. While mitigated with noise injection, it remains a limitation that needs further analysis.
>
> What the reviewer described matches our experience when training video generators for real-world tasks, where data is often scarce. This is a common issue in learning-based models trained with insufficient data, where the model tends to over-fit to spurious features. While scaling up the dataset can help mitigate this problem, collecting sufficient in-domain data in robotics is often difficult. Future work could explore architectural designs that better disentangle the model’s internal belief inference from the video generation objective, or pre-train the video generator on in-the-wild videos to leverage general visual knowledge, which we believe is promising given the abundance of video data available online.

---

> ### Author Response · Authors · 2025-11-28
> **Response to reviewer DZm1 (2/2)**
>
> ### Require the whole dataset for retrieval-based estimation.
> > The retrieval-based state estimation requires storing and accessing the entire experience dataset at test time, which may not be scalable or efficient for long-term deployment compared to a fully learned belief state. No proper discussion on this problem has been conducted and analyzed.
>
> We want to respectfully point out that we have already discussed this scenario in our main experiments. In section 5.3, we introduced the Ours (Refine) baseline, which obtains the state embedding by running the refinement process on randomly sampled embeddings. Since this baseline omits the retrieval process, it suits the scenario the reviewer described. While it underperformed our full pipeline, it still consistently outperforms other baselines, showing its effectiveness.
>
>
> ### Clarify the notation of “planning.”
> > Last, the notion of planning used in the paper is not the classical one used, for instance, in the AI Planning community, and it seems more related to motion planning. The authors shall better characterize their notion of planning, and the relation that may exist with POMDP planning (i.e. synthesize a policy to achieve a goal in a partially observable Markov decision problem).
>
> We thank the reviewer for suggesting this comparison. We have clarified our use of the term “planning” and explicitly distinguished it from AI planning and POMDP planning in the Related Work section to make the contrast clearer.
>
>
> ### Reproducibility
> > The additional material, consisting of just a link to another web page, is not informative enough to enable an individual to reproduce the paper's results.
>
> We thank the reviewer for raising this concern. We provided all relevant materials, including a link to the codebase for reproducing the results, on the project webpage at the time of submission. We have now further clarified this by adding the code link directly in the paper.

---

### Official Review · Reviewer_1k5g · 2025-10-27

**Soundness:** 2
**Presentation:** 2
**Contribution:** 2
**Rating:** 4
**Confidence:** 3

**Summary:**

This paper proposes Implicit State Estimation via Video Replanning, a framework that enables robots to adapt plans during execution by leveraging failed interactions without explicitly modeling system parameters. The system maintains buffers of failed plans and interactions, updates latent state embeddings online, and uses a video diffusion model to generate improved future plans. A novel Meta-World System Identification Benchmark is introduced to evaluate adaptation across tasks with hidden physical parameters. Experiments in simulation and real-world setups demonstrate improved replanning efficiency and robustness compared to video planning, imitation learning, and offline RL baselines.

**Strengths:**

strengths

1: Novel implicit adaptation mechanism that integrates past failures into video-based planning without requiring explicit parameter estimation or belief models.

2: Comprehensive evaluation, including a new benchmark, ablations, and real-world experiments, showing strong performance and practical feasibility.

3: Modular and general framework that enhances existing video planning pipelines and leverages diffusion models for flexible plan generation.

**Weaknesses:**

Weaknesses

1: Computational overhead: Online embedding refinement and multiple video plan generations increase inference cost, limiting scalability in real-time or resource-constrained settings.

2: Limited theoretical analysis: While effective empirically, the method lacks formal guarantees on convergence, stability, or conditions under which implicit state estimation succeeds.

3: Dependence on visual similarity and pixel-space distances for plan rejection may struggle in cluttered scenes or visually ambiguous scenarios.

**Questions:**

1: Scalability of Implicit State Embedding:
How does the proposed framework scale when applied to more complex environments with higher-dimensional hidden state variables or long-horizon tasks? Is there any evidence that implicit state embeddings remain reliable as task complexity increases?

2: Generalizability Across Objects and Tasks:
The method relies on retrieval from past interaction data grouped by object ID, how well does the approach generalize to entirely new objects or tasks without prior failure data? Would the system fail or fall back to vanilla video planning in such cases?

3: Efficiency vs. Computational Cost:
Given that the framework involves repeated video generation, embedding refinement, and rejection-based search, what is the real-time inference cost? Is the method practical for deployment in time-sensitive robotic applications, and how does the latency compare to conventional planners or belief-state methods?

---

> ### Author Response · Authors · 2025-11-28
> **Response to reviewer 1k5g (Part 1/3)**
>
> We thank the reviewer for raising their concerns. Below, we address each question in detail.
>
> ### Efficiency/Computational Cost
> > Online embedding refinement and multiple video plan generations increase inference cost, limiting scalability in real-time or resource-constrained settings.
> Given that the framework involves repeated video generation, embedding refinement, and rejection-based search, what is the real-time inference cost? Is the method practical for deployment in time-sensitive robotic applications, and how does the latency compare to conventional planners or belief-state methods?
>
> First, we emphasize that our video planner replans once per episode, not once per control step. This matches the setting of conventional latent-belief methods such as HiP-MDP [1] and PEARL [2]. After a belief or plan is computed, both these methods and ours use a lightweight policy to produce actions, which incurs negligible latency and is therefore practical for real-time deployment.
>
> In fact, existing baselines also require multiple model evaluations to compute reliable beliefs. Specifically, PEARL aggregates outputs from multiple state-transition tuples to infer the latent context, while HiP-MDP performs test-time optimization that typically involves around 100 gradient steps per belief estimation, which is comparable to our refinement stage in runtime.
>
> For applications that are extremely latency-sensitive, such as dynamic locomotion, one may instead rely on regression-style estimators such as RMA [3]. However, such methods typically assume access to precise simulation environments with ground-truth system parameters and highly accurate dynamics models. Our setting differs in that it targets visually grounded and uncertain real-world conditions, where such supervision is unavailable.
>
> Importantly, higher per-replan cost does not imply worse deployment performance. A more meaningful metric in practice is wall-clock time until the first successful execution, rather than latency per planning step. In manipulation tasks, runtime is typically dominated by physical execution rather than planning. As shown in the table below, although our method incurs a higher planning cost per replan, it requires substantially fewer replans to succeed, resulting in the lowest average wall-clock time to success. This advantage is further amplified in realistic settings where environment resets are costly.
>
> Finally, with improved hardware and faster algorithms, our advantage in time-to-success is expected to further increase, since real-world execution time remains fixed while planning continues to benefit from computation advances. For reference, all reported timings were measured on an RTX 3080 Ti GPU, which is already exceeded by many modern research labs.
>
> |                 | AVDC     | PEARL    | HiP-MDP | Ours     |
> | --------        | -------- | ------   |-------- | ---      |
> | replan cost (s) | 2.3      | 1.9      | 4.8     | 16.5     |
> | execution cost/replan (s)  |25 |25    |25|25|
> | total cost/replan (s)| 27.3 | 26.1 | 29.8 | 41.5 |
> | replans/success | 6.83     | 8.64     | 8.65    | 4.26     |
>  success cost (s)| 186.46|225.504|257.77|**176.79**|
>
> ### Rejection strategy in visually challenging scenarios
> > Dependence on visual similarity and pixel-space distances for plan rejection may struggle in cluttered scenes or visually ambiguous scenarios
>
> We agree with the reviewer that pixel-level distances may struggle in cluttered or visually ambiguous scenarios. One natural source of difficulty in such settings is the presence of visual signals that are unrelated to the underlying interaction dynamics. To evaluate this factor in isolation, we introduce two environment variants with randomized backgrounds that simulate task-irrelevant visual distractors.
>
> **Table 3: Distance Function Performance under Noisy Environments.**
> Metric: Replans Until Success
> |               | Slide Brick Noisy | Open Box Noisy |
> | --------      | --------    | -------- |
> | No Rejection  | 6.09        | 1.91     |
> | DINO          | 5.56        | 1.65     |
> | L2 error      | 4.89        | 1.79     |
> | Seg + DINO    | 5.54        | **1.65** |
> | Seg + L2 error| **4.53**    | **1.65** |
>
> As shown in Table 3, under these noisy conditions, L2-based rejection remains competitive and continues to outperform DINO on the Slide Brick task, suggesting that fine-grained pixel-level cues can still be informative for estimating hidden parameters in interaction-centric tasks. While DINO performs slightly better on Open Box, both metrics outperform the no-rejection baseline, indicating that rejection remains effective regardless of the underlying distance metric, highlighting its robustness. For more details about this experiment, please refer to Appendix D.7.

---

> ### Author Response · Authors · 2025-11-28
> **Response to reviewer 1k5g (Part 2/3)**
>
> ### Theoretical analysis
> > While effective empirically, the method lacks formal guarantees on convergence, stability, or the conditions under which implicit state estimation succeeds.
>
> We thank the reviewer for urging us to include formal analysis for our approach. To address the reviewer's concern, we provide a theoretical justification of our approach below:
>
> **Preliminaries.**
> Let
> $$
> \mathcal{K} = \{1, \dots, K\}
> $$
> denote a finite set of latent interaction modes, and let
> $$
> q = (q_1, \dots, q_K), \quad q_k = \Pr(\text{mode } k)
> $$
> be the planner’s belief distribution.
> Let the correct mode be $k^\star$, with probability
> $$
> q_{k^\star} \in (0,1).
> $$
>
> At each replanning step, the planner draws $N$ independent samples
> $$
> K_1, \dots, K_N \sim q,
> $$
> generates predicted outcomes and selects the one that is farthest from the previously failed execution.
>
> **Rejection as probability amplification.**
> We assume a mild separation condition: rollouts produced by the same failing mode remain closer to the failed execution than those produced by other modes.
> Under this assumption, rejection selects the correct mode whenever $k^\star$ appears at least once among the $N$ samples.
>
> The probability that at least one of the $N$ samples equals $k^\star$ is
> $$
> p_{\mathrm{rej}} = 1 - (1 - q_{k^\star})^N.
> $$
> For any $N \ge 2$, we have $p_{\mathrm{rej}} > q_{k^\star}$ whenever $q_{k^\star} \in (0,1)$, showing that rejection strictly amplifies the probability of selecting the correct interaction mode.
>
> **Convergence under replanning.**
> Each replanning attempt succeeds with probability at least $p_{\mathrm{rej}}$.
> Assuming independence across attempts, the probability that no success is observed after $M$ replans is
> $$
> \Pr(\text{no success after } M) = (1 - p_{\mathrm{rej}})^M.
> $$
> Using the standard inequality
> $$
> 1 - x \le e^{-x}, \quad \text{for all } x \in [0,1],
> $$
> we obtain the exponential bound
> $$
> \Pr(\text{no success after } M) \le e^{-p_{\mathrm{rej}} M}.
> $$
>
> This shows that the failure probability decays exponentially with the number of replans.
> Moreover, since $p_{\mathrm{rej}}$ is a monotonically increasing function of $N$, sampling more hypotheses per step further accelerates convergence.
>
>
> ### Scalability of Implicit State Embedding
> > How does the proposed framework scale when applied to more complex environments with higher-dimensional hidden state variables or long-horizon tasks? Is there any evidence that implicit state embeddings remain reliable as task complexity increases?
>
> To study how the proposed embedding scales to more complex settings, we construct a harder multi-faucet environment in which the robot must sequentially manipulate three faucets instead of one. Each faucet can only be turned when pushed from the correct side, and pushing from the wrong side causes the handle to jam. This leads to 8 distinct interaction modes and a longer planning horizon, making both the hidden parameter space and the decision process substantially more complex than in the original task.
>
> We use this environment to evaluate whether implicit state embeddings can still represent the underlying system configuration accurately. Specifically, we extract embeddings from datasets of different sizes and then apply our retrieval procedure on new interaction trials, following the same pipeline as in the main paper. We measure performance by comparing the retrieved system configuration against the ground-truth hidden parameters and report the success rate of exact matches.
>
> **Table 4: Hidden Parameter Matching Performance of Our State Embeddings.**
> Metric: Hidden Parameter Matching Success Rate (%)
>
> | Method \ Dataset Size (Success, Fail)  | (2, 14) | (10, 70) | (40, 280) | (120, 840) |
> | -------- | -------- | --- | --- | --- |
> | Random | 11.87 +- 1.28  | 11.87 +- 1.28  | 11.87 +- 1.28  | 11.87 +- 1.28      |
> | **Ours (Retrieval)** | **18.59 +- 1.54** | **30.94 +- 1.83** | **53.12 +- 1.97** |**67.66 +- 1.85**  |
>
> As shown in Table 4, our retrieval-based method consistently outperforms the baseline across all dataset sizes. Even with very limited experience, the embeddings capture meaningful structure in the system dynamics and allow meaningful recovery of the correct interaction mode. As more data becomes available, performance improves steadily and reaches over 67% exact recovery with the largest dataset.
>
> These results indicate that the embedding does not collapse as task complexity increases. Instead, it continues to organize interactions in a way that separates different hidden configurations, proving the scalability of the proposed implicit state representation beyond single-object, short-horizon tasks. Please refer to Appendix D.5 for more details about this experiment.

---

> ### Author Response · Authors · 2025-11-28
> **Response to reviewer 1k5g (Part 3/3)**
>
> ### Generalizability Across Objects and Tasks
> > The method relies on retrieval from past interaction data grouped by object ID, how well does the approach generalize to entirely new objects or tasks without prior failure data? Would the system fail or fall back to vanilla video planning in such cases?
>
> We want to take this chance to clarify that, although our state embeddings are indexed by object ID, this does not imply that our approach is restricted to memorizing specific objects. Each embedding summarizes the interaction dynamics induced by a particular physical configuration, rather than surface appearance. The video encoder and retrieval process are therefore driven by how the environment responds to action, not by visual identity.
>
> As a result, when a novel object at test time exhibits interaction patterns similar to those seen in training, the system can retrieve a relevant canonical embedding and apply appropriate behavior, even if that object has never been observed before. In practice, retrieval operates over latent dynamics, not object identity, enabling transfer across previously unseen instances that share similar physical properties.
>
> To further evaluate generalizability, we construct novel environments where object appearances are different from training, but the underlying interaction mechanism remains the same. For each stored state embedding, we record the ground-truth system parameters it represents for evaluation purposes. During evaluation, we compare the system parameters of the retrieved embedding with those of the current environment in which the novel object appears and report the error. We consider the following baselines:
>
> 1. Random: Randomly select one embedding.
> 2. ISE-Seen: Retrieval using the ISE-Retrieval approach on seen environments.
> 3. ISE-Unseen: Retrieval using the ISE-Retrieval approach on novel environments.
>
> **Table 5: Generalization to Unseen Instances.**
> Metric: Mode/hidden parameter retrieval error.
>
> | Method \ Task | push_bar | open_box | pick_bar | turn_faucet | slide_brick |
> |----------------|------------|------------|------------|------------|------------|
> | Random | 0.0954 ± 0.0052 | 0.4900 ± 0.0500 | 0.0954 ± 0.0052 | 0.4900 ± 0.0500 | 0.0528 ± 0.0040 |
> | ISE-Seen | 0.0694 ± 0.0057 | 0.0200 ± 0.0140 | 0.0793 ± 0.0069 | 0.0200 ± 0.0140 | 0.0448 ± 0.0032 |
> | ISE-Unseen | 0.0693 ± 0.0049 | 0.0700 ± 0.0255 | 0.0729 ± 0.0054 | 0.0200 ± 0.0140 | 0.0428 ± 0.0032 |
>
> **Table 6: Normalized Performance.**
> *Random* is linearly normalized to 0 and *ISE Seen* is normalized to 1. The normalized values indicate how much performance is retained when interacting with novel object instances.
> |  | push_bar | open_box | pick_bar | turn_faucet | slide_brick | **Average** |
> |----------------|------------|------------|------------|------------|------------| --- |
> | Random | 0.0000 | 0.0000 | 0.0000 | 0.0000 | 0.0000 | 0.0000 |
> | ISE-Seen | 1.0000 | 1.0000 | 1.0000 | 1.0000 | 1.0000 | 1.0000 |
> | ISE-Unseen | 1.0028 | 0.8936 | 1.3971 | 1.0000 | 1.2468 | 1.1081 |
>
> The experiment confirms that our method remains robust when facing novel object instances. Even when object appearance changes, the system is able to retrieve the correct interaction mechanism from previously seen environments. This shows that the retrieved embeddings capture transferable structure rather than object-specific details, allowing the model to apply prior knowledge effectively in new settings. For more details about this experiment, please refer to Appendix D.8.
>
> ### References
> [1] Killian, Taylor W. et al. “Robust and Efficient Transfer Learning with Hidden Parameter Markov Decision Processes.” Advances in neural information processing systems 30 (2017): 6250-6261 .
>
> [2] Rakelly, Kate et al. “Efficient Off-Policy Meta-Reinforcement Learning via Probabilistic Context Variables.” International Conference on Machine Learning (2019).
>
> [3] Kumar, Ashish et al. “RMA: Rapid Motor Adaptation for Legged Robots.” RSS (2021).

---

### Official Review · Reviewer_u999 · 2025-10-28

**Soundness:** 3
**Presentation:** 2
**Contribution:** 2
**Rating:** 4
**Confidence:** 4

**Summary:**

Summary

The paper introduces a new framework for adaptive video-based decision-making in robotics. The system uses an adaptive video planning framework, wherein a video generation model attempts to generate candidate future rollouts conditioned on the current scene and a hypothesized latent state embedding, which are then converted into actions.  The key innovation is that the planner can adapt its behavior base on previous failed trials.  When a robot fails during task execution (e.g., pushing instead of pulling a door), the failure itself contains valuable information about hidden environment parameters—like friction, mass, or interaction mode.  The planner uses two forms of adaptation:
- implicit state estimation (ISE): The system refines its internal latent “state embedding” by minimizing the discrepancy between its generated rollouts and the observed interaction.  This allows it to make more realistic video generations for the next trial.
- Rejection-based replanning: at each planning round, the Video Plan Generator generates N candidate plans, covering a range of possible outcomes.  To prevent the agent from repeatedly attempting failed plans, the plan that has the maximum L2 pixel distance from previous failed plans is selected.

**Strengths:**

- The problem formulation is clear and well-motivated.  The problem is important to the field.  I especially liked the example given in the introduction of opening a door without knowing whether it should be pushed or pulled.  This made it really clear what the conceptual challenge was.
- The approach is interesting.
- The experimental domains seem reasonably interesting/challenging.
- The paper proposes a new benchmark (Meta-World System Identification Benchmark) with 5 manipulation-style tasks where key parameters (e.g. center of mass, friction, push vs pull mode) are hidden and must be inferred through interaction, which feels like a good testbed for this setting.
- Empirically, the method outperforms AVDC, BC, and offline RL (CQL) in terms of ‘replans needed until success,’ suggesting that leveraging failed attempts is actually useful in practice.

**Weaknesses:**

There were many parts of the paper that I found lacked sufficient detail.
- In Sec. 4.1, in the Representing state embeddings section, it is stated that “we preprocess the dataset by grouping entries by object ID and selecting a successful trial as the canonical embedding e^o_j”.  I don’t understand what these “canonical embeddings” are or how they are used, as the paper never explicitly revisits how these canonical embeddings are stored, updated, or surfaced at inference time, which made it hard to connect the preprocessing step to the online retrieval procedure in Sec. 4.1.
- In Sec. 4.1, in the Retrieval and refinement section, it is unclear what is meant by the “current video”.  If we haven’t yet performed an interaction, how do we have a current video?  Is this an initial frame, or some initial sequence of frames?
- What are state embeddings?  Is this a vector that we get from a video, a vector for each frame, or something else?  This is central and it should really be formalized.
- Figure 3 does not illustrate refinement, as far as I can tell, which is a shame because I found the written text explanation very vague.
- The approach could really benefit from more mathematical descriptions of what’s going on.  The text descriptions tend to be vague and hard to follow.
- Dataset preprocessing: what is object ID?  Is this just the object being manipulated, or is it also the “scenario” as well?  I.e., in the faucet example, are trials where the faucet moves in the same way grouped together?
- I don’t see it clearly stated what action space is used for the baselines.
- The ablation labels in Fig. 5 (AVDC+Retrieval, AVDC+Rejection, Ours) aren’t described in enough procedural detail to be reproducible. For example, what exactly is being fed to the planner in AVDC+Retrieval vs AVDC? Is AVDC+Retrieval simply AVDC but conditioned on a single retrieved embedding instead of no embedding? This should be clarified.


There are also some conceptual weaknesses:
- The problem setting is not novel, as far as I can tell it is the same one studied in Hidden-Parameter MDPs [1].  Many approaches exist for rapidly adapting in environments with the same type of uncertainty studied here [2,3,4].  The refinement method used in this paper seems to be a fairly heuristic solution, approximating what some of these other papers do in a more principled manner.  From a formulation standpoint, this is extremely close to Hidden-Parameter MDPs and latent-context meta-RL, except the latent context is represented by a vision encoder + diffusion refinement, and the policy is instantiated via video plan + tracking instead of a standard parametric policy. The paper does not compare against prior latent-context adaptation methods, so I cannot tell if this buy-in to video planning actually provides an advantage beyond aesthetics.
- an 8-frame video plan seems extremely short.  Does this really mean that the approach can only plan 7 timesteps into the future?  What is the control frequency? Are these 7 steps high-level waypoints that get tracked over longer horizons via the Action Module, or is this genuinely planning only ~7 control steps ahead?
- With regard to the rejection strategy, L2 distance in pixel space seems like it would be a poor distance metric, as it doesn’t really capture any semantic detail about what’s actually happening in the plan, and is sensitive to things like subtle shifts in camera view. The authors do provide an ablation (Table 4 / Fig. 6) showing that pixel distance actually outperforms DINOv2 feature distance on their benchmark, which partially mitigates this concern, but I still worry about generalization to more realistic visual variability.
- Using a set of heuristic policies for the action module seems somewhat limited and hard-coded.
- Because I don’t know what action space is being used for the baselines, I can’t tell if a fair comparison is being done.
- Is there no ablation that keeps only the refinement component but removes the rejection component?  This seems like a major oversight as refinement is one of the main proposed methods of adaptation.


[1] Doshi-Velez, Finale, and George Konidaris. "Hidden parameter markov decision processes: A semiparametric regression approach for discovering latent task parametrizations." IJCAI: proceedings of the conference. Vol. 2016. 2016.
[2] Rakelly, K., Zhou, A., Finn, C., Levine, S. &amp; Quillen, D.. (2019). Efficient Off-Policy Meta-Reinforcement Learning via Probabilistic Context Variables. <i>Proceedings of the 36th International Conference on Machine Learning</i>, in <i>Proceedings of Machine Learning Research</i> 97:5331-5340 Available from https://proceedings.mlr.press/v97/rakelly19a.html
[3] Sæmundsson, Steindór, Katja Hofmann, and Marc Peter Deisenroth. "Meta reinforcement learning with latent variable gaussian processes." arXiv preprint arXiv:1803.07551 (2018).
[4] Killian, Taylor W., et al. "Robust and efficient transfer learning with hidden parameter markov decision processes." Advances in neural information processing systems 30 (2017).

**Questions:**

Please clarify the questions I raised in the weaknesses section.

---

> ### Author Response · Authors · 2025-11-28
> **Response to reviewer u999 (Part 1/3)**
>
> We sincerely thank the reviewer for the constructive feedback. Below, we address each question in detail.
>
> ### Clarification
>
> ***What are state embeddings?***
>
> A *state embedding* in our framework is a single, fixed-dimensional vector computed from an interaction video rather than from individual frames. Each video is subsampled into \(F = 8\) frames, each frame is encoded using a pretrained vision encoder, and the resulting features are aggregated into a single video-level representation.
>
> Importantly, state embeddings represent object behavior rather than static appearance. Because they are constructed from interaction videos, they capture latent physical properties such as friction, articulation direction, or functional constraints that are not observable from a single image.
>
> We have revised Sec. 4.1 to explicitly define state embeddings and describe how they are constructed and used in the pipeline.
>
> ***What is an object ID? Does it correspond to the object instance only, or also the scenario?***
>
> In our framework, an *object ID* is not a visual category label, but an index for a fixed hidden configuration used during data collection.
>
> Concretely, object IDs are assigned by the data collection protocol: whenever we know we are interacting with the same physical configuration (same hidden system parameters), all interaction videos collected under that setup share the same object ID. We do **not** attempt to infer configuration equality automatically from observations; instead, object ID is used as a one-way indicator:
> - if two videos share the same object ID, they are guaranteed to come from the same underlying configuration;
> - if they have different object IDs, we do not assume whether their parameters are similar or different.
>
> As an illustration, in an environment containing multiple components (e.g., several faucets), as long as the physical configuration remains fixed, all interactions collected in that environment are assigned the same object ID. The ID refers to the shared hidden configuration that governs the interaction dynamics, not to individual geometric parts.
>
> Since our benchmarks focus on single-object manipulation per task, we use “object ID” to index physical variation in an intuitive way, rather than using more abstract naming.
>
> ***Regarding “canonical embeddings”:***
>
> We clarify that *canonical embeddings* are not separate from state embeddings, but a specific subset of them. In our framework, we distinguish between:
>
> 1. **Video-level state embeddings**: embeddings computed from individual interaction videos, and
> 2. **Canonical embeddings**: one selected embedding per object ID that serves as the object’s representative state.
>
> Concretely, for each object ID $o$, we compute embeddings for all its interaction videos. Among them, we select one successful trial and store its embedding as the *canonical embedding* $e^o$. This canonical embedding is fixed during training and stored in a retrieval database.
>
> At inference time, a new interaction video is encoded into a query embedding, which is used to retrieve the nearest canonical embedding. The retrieved canonical embedding is then used as the object-level state representation for subsequent planning.
>
> We have revised Sec. 4.1 to make this distinction explicit and to clarify that the offline preprocessing step constructs the retrieval database, while the online procedure performs video-level retrieval followed by object-level lookup.
>
> ***Regarding the meaning of “current video” in Sec. 4.1:***
>
> In our system, retrieval and refinement operate only **after** the agent has collected its first interaction video in the current episode. Before any interaction occurs, there is no “current video,” and thus retrieval is naturally skipped. In this initial stage, the planner conditions on the first visual observation (the initial frame) together with a learned "null state embedding", which represents the “no-interaction-yet” condition. We have clarified this point in the revised manuscript.
>
> ***Action space***
> All baselines in our experiments use the *same* action space as our method to ensure a fair comparison. Specifically, we adopt a 4-dimensional positional control action space $(dx, dy, dz)$,  where $(dx, dy, dz)$ are Cartesian end-effector displacements and $g$ controls the gripper. We have updated the Experiment section to state this explicitly in the experimental setup.
>
> ***Plan horizon***
> The 8-frame video plan indeed provides high-level waypoints that span the **entire episode**. The Action Module tracks the visual trajectory between these waypoints over many low-level control steps.
> In other words, the planning horizon is not 7 control steps: the video plan specifies a coarse visual trajectory for the full task, while low-level control runs at a higher frequency (20Hz) to follow this trajectory throughout the episode. We have clarified this in the paper.

---

> ### Author Response · Authors · 2025-11-28
> **Response to reviewer u999 (Part 2/3)**
>
> ### Figure descriptions
>
> We thank the reviewer for the helpful feedback on the figure descriptions. We have made the following revisions in the paper to improve clarity.
>
> ***Refinement in Figure 3***
> > Figure 3 does not illustrate refinement, as far as I can tell, which is a shame because I found the written text explanation very vague.
>
> The caption of Figure 3 now states that the Retrieval Module includes both the retrieval and refinement processes.
>
> ***Data input for methods in Fig. 5***
> > The ablation labels in Fig. 5 (AVDC+Retrieval, AVDC+Rejection, Ours) aren’t described in enough procedural detail to be reproducible. For example, what exactly is being fed to the planner in AVDC+Retrieval vs AVDC? Is AVDC+Retrieval simply AVDC but conditioned on a single retrieved embedding instead of no embedding? This should be clarified.
>
> We added detailed descriptions of the inputs used by each method in Figure 5.
>
> ### More mathematical method descriptions
> We thank the reviewer for the helpful suggestion to include more mathematical formulations. We have updated the paper with explicit definitions for the state embeddings and the retrieval and refinement procedure, and we believe this improves the overall clarity of the presentation.
>
> ### Problem setting against prior latent-context adaptation methods
> > The problem setting is not novel, as far as I can tell it is the same one studied in Hidden-Parameter MDPs [1]. Many approaches exist for rapidly adapting in environments with the same type of uncertainty studied here [2,3,4]. The refinement method used in this paper seems to be a fairly heuristic solution, approximating what some of these other papers do in a more principled manner. From a formulation standpoint, this is extremely close to Hidden-Parameter MDPs and latent-context meta-RL, except the latent context is represented by a vision encoder + diffusion refinement, and the policy is instantiated via video plan + tracking instead of a standard parametric policy. The paper does not compare against prior latent-context adaptation methods, so I cannot tell if this buy-in to video planning actually provides an advantage beyond aesthetics.
>
> We agree with the reviewer that our problem setting is related to Hidden-Parameter MDPs and latent-context adaptation. Therefore, we have added a new paragraph in Related Works to discuss these works. To furthur address the reviewer's concern, we extended our evaluation to include representative methods from this line of work, specifically HiP-MDP [1] and PEARL [2], under the same settings of our main experiment.
>
> We implemented both methods as latent-context conditioned actor–critic policies (HiP-MDP-AC and PEARL-AC) and evaluated them directly against our approach. Notably, these baselines rely on privileged information during training, including dense reward functions and action supervision, which are not available to our method. In addition, we integrated the latent-state estimation models trained by these approaches directly into our framework by replacing our Retrieval Module with their inferred embeddings, while keeping the rest of the pipeline unchanged (HiP-MDP-ISE and PEARL-ISE). This demonstrates that prior state estimation approaches can be seamlessly integrated into our system, benefiting from our refinement and rejection procedures.
>
> The results support two conclusions. First, our full video-planning system consistently outperforms these latent-adaptation baselines, even when those methods are given access to privileged supervision. Second, incorporating external embeddings into our ISE framework improves performance over the original actor–critic implementations, showing that our refinement and rejection mechanisms remain effective across different embedding spaces.
>
> Together, these results suggest that the integrated formulation in our framework empirically performs better than prior latent-state adaptation methods, and that the improvements are driven by the combined planning and adaptation mechanism rather than a particular embedding choice.
>
> **Table 1: Comparison with Latent-Context Adaptation Methods.**
> Metric: Replans Until Success
> | Method | Push Bar | Pick Bar | Slide Brick | Open Box | Turn Faucet  | Overall |
> |---|---|---|---|---|---|---|
> | HiP-MDP-AC   | 8.65 ± 0.24    | 7.65 ± 0.23    | 11.29 ± 0.22    | 13.01 ± 0.15   | 13.03 ± 0.14    | 5.10 ± 0.07 |
> | PEARL-AC     | 8.64 ± 0.23    | 8.99 ± 0.24    | 13.20 ± 0.13    | 13.09 ± 0.14   | 2.57 ± 0.09     | 3.73 ± 0.06 |
> | HiP-MDP-ISE  | 7.88 ± 0.24    | 9.62 ± 0.23    | 10.35 ± 0.23    | 13.21 ± 0.13   | 13.06 ± 0.14    | 5.17 ± 0.07 |
> | PEARL-ISE    | 5.73 ± 0.21    | 5.70 ± 0.21    | 7.69 ± 0.23     | 1.60 ± 0.04    | 1.60 ± 0.05     | 1.28 ± 0.04 |
> | **Ours**     | **4.26 ± 0.20** | **3.84 ± 0.18** | **6.69 ± 0.26** | **1.25 ± 0.10** | **1.39 ± 0.14** | **1.00 ± 0.03** |

---

> ### Author Response · Authors · 2025-11-28
> **Response to reviewer u999 (Part 3/3)**
>
> ### Generalizability for the L2 distance rejection strategy
>
> We agree with the reviewer that pixel-level L2 distance does not explicitly encode semantic similarity and can be sensitive to task-irrelevant visual variation. To directly address this concern, we introduce two environment variants with randomized background distractors to simulate what commonly appear in more realistic settings.
>
> **Table 2: Distance Function Performance under Noisy Environments.**
> Metric: Replans Until Success
> |               | Slide Brick Noisy | Open Box Noisy |
> | --------      | --------    | -------- |
> | No Rejection  | 6.09        | 1.91     |
> | DINO          | 5.56        | 1.65     |
> | L2 error      | 4.89        | 1.79     |
> | Seg + DINO    | 5.54        | **1.65** |
> | Seg + L2 error| **4.53**    | **1.65** |
>
> As shown in Table 2, under these noisy conditions, L2-based rejection remains competitive and continues to outperform DINO on the Slide Brick task, suggesting that fine-grained pixel-level cues can still be informative for estimating hidden parameters in interaction-centric tasks. While DINO performs slightly better on Open Box, both metrics outperform the no-rejection baseline, indicating that rejection remains beneficial regardless of the underlying distance metric.
>
> We further demonstrate a simple and practical mitigation strategy by incorporating segmentation to isolate task-relevant regions. When background pixels are masked out, both pixel-based and feature-based distances become more robust, and segmentation + L2 achieves the best overall performance. Importantly, this remedy is feasible in practice: high-quality segmentation masks can now be obtained from modern vision–language models such as Segment Anything Models (SAM), making object-centric distance evaluation realistic in visually complex environments.
>
> Regarding camera sensitivity, potential mitigations include using pose estimation to track object pose relative to the robot or applying standard video stabilization techniques prior to distance computation. We view contructing distance functions that are robust to viewpoint changes as an important research direction but out-of-scope for this work. It would broadly benefit many vision-based control and adaptation systems beyond our setting.
>
> ### Use of heuristic policies
>
> We note that the use of heuristic controllers is common in prior video-based planning frameworks, either as a direct executor [3,4] or as guidance for policy learning [5]. In this work, our primary focus is the video-planning component, specifically how to generate and select plans that adapt effectively to unknown environment dynamics, rather than advancing execution policies themselves. Importantly, our framework does not depend on a particular controller design. The Action Module can be replaced by alternative video-based policy designs such as inverse-dynamics models [6] or learned goal-conditioned [7] or trajectory-conditioned [8] policies. We believe that improving video-based policy is orthogonal to the central contribution of this work, and we are actively investigating more advanced controllers in our follow-up work.
>
> ### References
>
> [1] Killian, Taylor W. et al. “Robust and Efficient Transfer Learning with Hidden Parameter Markov Decision Processes.” Advances in neural information processing systems 30 (2017): 6250-6261 .
>
> [2] Rakelly, Kate et al. “Efficient Off-Policy Meta-Reinforcement Learning via Probabilistic Context Variables.” International Conference on Machine Learning (2019).
>
> [3] Yuan, Chengbo et al. “General Flow as Foundation Affordance for Scalable Robot Learning.” Conference on Robot Learning (2024).
>
> [4] Ren, Juntao et al. “Motion Tracks: A Unified Representation for Human-Robot Transfer in Few-Shot Imitation Learning.” 2025 IEEE International Conference on Robotics and Automation (ICRA) (2025): 8802-8810.
>
> [5] Bharadhwaj, Homanga et al. “Track2Act: Predicting Point Tracks from Internet Videos Enables Generalizable Robot Manipulation.” European Conference on Computer Vision (2024).
>
> [6] Du, Yilun et al. “Learning Universal Policies via Text-Guided Video Generation.” Neural Information Processing Systems (2023)
>
> [7] Luo, Yunhao and Yilun Du. “Grounding Video Models to Actions through Goal Conditioned Exploration.” International Conference on Learning Representations (2024).
>
> [8]Fang, Xiaolin et al. “KALM: Keypoint Abstraction Using Large Models for Object-Relative Imitation Learning.” 2025 IEEE International Conference on Robotics and Automation (ICRA) (2024): 8307-8314.

---

### Author Response · Authors · 2025-12-02
**To Area Chair: Summary of our rebuttal**

Dear Area Chair,

We sincerely appreciate the time and effort dedicated by all reviewers. We are encouraged by their recognition of the relevance and potential impact of our work, particularly in the following aspects:

**Problem & Motivation.**
Reviewers acknowledged the importance of adaptive planning under hidden system parameters and the relevance of implicit state estimation for robot interaction (u999, 1k5g, DZm1). The proposed direction was viewed as meaningful for bridging video-based planning and online adaptation (1Eqz).

**Methodology & Empirical Evaluation.**
The framework was recognized as a coherent combination of retrieval, refinement, and replanning (u999, 1k5g), and the experimental coverage across simulation and real-world settings was viewed as comprehensive (1k5g, 1Eqz). Reviewers also noted the promising performance gains in replans-until-success across tasks (u999, DZm1).


---

We performed additional analysis, comparisons, and experiments to address the reviewers’ concerns directly. All revisions in the manuscript are marked in blue. Key updates include:

- **Clarification of state embeddings and formulation (u999, 1Eqz).**
In response to questions on what state embeddings represent, how canonical embeddings are constructed, and how “current video” and object ID are defined (u999), and on the distinction between plans and interactions and the meaning of “implicit” estimation (1Eqz), we explicitly defined embeddings as video-level representations of latent physical properties, clarified object ID semantics, and refined the formal description of retrieval, refinement, and estimation.

- **Theoretical analysis of the rejection mechanism (1k5g).**
To address concerns about the lack of theoretical grounding (1k5g), we added a probabilistic analysis showing that rejection amplifies the probability of selecting the correct interaction mode and that failure probability decays exponentially under replanning.

- **Efficiency and deployment concerns (1k5g, DZm1).**
In response to questions about real-time feasibility and compute cost (1k5g, DZm1), we added runtime breakdowns, clarified that replanning occurs once per episode, and showed that despite higher per-replan cost, our method achieves lower expected time-to-success than baselines.

- **Robustness under visual noise (u999, 1k5g).**
To address concerns about pixel-based rejection in cluttered scenes and visual variability (u999, 1k5g), we introduced noisy-background environments, evaluated multiple distance metrics, and demonstrated that rejection remains effective, especially when combined with segmentation.

- **Generalization and scalability (1k5g, 1Eqz, DZm1).**
In response to questions about scaling to more complex systems and longer horizons (1k5g), generalization to unseen objects and tasks (1Eqz), and long-term applicability of retrieval-based estimation (DZm1), we added a harder multi-faucet task. We performed generalization experiments showing improved retrieval accuracy with data scale and transfer driven by interaction dynamics.

- **Additional baselines and retrieval variants (u999, 1Eqz).**
To include comparisons with latent-context-based methods (u999) and additional retrieval-style baselines and alternatives to rejection (1Eqz), we added latent-context adaptation baselines (HiP-MDP and PEARL) and evaluated attraction-based selection.

- **Benchmark framing, data collection, and reproducibility (1Eqz, DZm1).**
In response to concerns about benchmark positioning and reproducibility (1Eqz, DZm1), we revised benchmark wording, clarified data collection procedures, and made the code link explicit.

We believe the rebuttal has addressed all concerns to the best of our ability through additional analysis, experiments, and clarifications. We respectfully submit this summary as a guide to the main improvements developed during discussion, and hope it is helpful for your assessment of the paper in its most complete form.


Finally, we sincerely thank you for your careful handling of our submission and for your commitment to maintaining the integrity of the review process.

Sincerely,

The Authors

---

### Meta-Review · Area_Chair_DTJ2 · 2026-01-07

**Summary:**

Reviewers agreed that the paper tackles a well motivated problem and praised the clarity of the problem formulation. Common concerns included insufficient methodological detail in the original submission, limited theoretical justification for the rejection mechanism, questions about scalability to more complex tasks or unseen objects, and doubts about the robustness of pixel space plan rejection in visually cluttered scenes. In the end, based on the opinions of various reviewers, the paper did not meet the acceptance criteria.

**Reviewer Concerns:**

The authors provided thorough responses to all weaknesses and questions raised by the reviewers.

**Reviewer Scores:**

Reviewer u999 (initial 4) may raise or maintain their score given the extensive clarifications on embeddings, object ID semantics, and new comparisons with latent-context baselines.
Reviewer 1k5g (initial 4) may raise or maintain their score following the added theoretical analysis, noisy-environment experiments, and scalability results.
Reviewer DZm1 (initial 6) is unlikely to change their score significantly.
Reviewer 1Eqz (initial 6) may raise or maintain their score after the authors clarified terminology.

---

### Decision · Program_Chairs · 2026-01-26

Reject